# Broadly neutralizing nanobodies target a defined structural pivot site on the RSV fusion protein

Qianqian Wang [1,2,3,5], Xianliang Ke[4,5], Entao Li[1,2,3,5], Dongxiang Hong[2,3], Zekai Cheng[2,3], Hongxin Li[4], Jiachen Zhang[2,3], Tengchuan Jin[2,3], Rui Gong[4], Bo Shu [4✉] & Sandra Chiu [1,2,3✉]

## Abstract

**Respiratory syncytial virus (RSV) is a leading cause of acute lower respiratory tract infections in infants and elderly individuals. Although nirsevimab represents a recent breakthrough against RSV infection, the emergence of resistant variants highlight the need for additional antiviral strategies. Here, we report two nanobodies (Nbs), 1G9 and 1D8, that target the RSV fusion (F) protein and exhibit high neutralizing activity against RSV subtypes. In vivo, the Nbs-Fc demonstrated robust prophylactic and therapeutic efficacy. Cryo-electron microscopy revealed that 1G9 and 1D8 specifically engage a conformational pivot site within antigenic site IV, cross-linking the metastable heptad repeats B (HRB) and the conformationally stable domain II of the F protein. This interaction stabilizes the prefusion conformation and prevents the structural rearrangement required for membrane fusion. Notably, the binding residues are highly conserved across RSV subtypes, accounting for the broad-spectrum neutralization observed. Together, our findings identify a structurally conserved and functionally critical epitope on RSV F and highlight 1G9 and 1D8 as promising candidates for next-generation prophylactic and therapeutic interventions against RSV.**

**Keywords** Respiratory Syncytial Virus; RSV F; Nanobody; Prophylactic and Therapeutic
**Subject Categories** Immunology; Microbiology, Virology & Host Pathogen Interaction

## Introduction

Respiratory syncytial virus (RSV) is a major pathogen that causes respiratory tract infections in infants and the elderly (Li et al, 2019; Li et al, 2022b; Lu et al, 2024). The clinical manifestations of patients infected with RSV are diverse. Mild cases present with the rhinorrhea, nasal congestion and cough, whereas severe cases may present with high fever, respiratory distress, and even death (Suleiman-Martos et al, 2021). Global surveillance data indicate that children under the age of 5 are at high risk for RSV infection, with those under 2 years facing the greatest risk and bearing the heaviest disease burden (Jiang et al, 2023). In 2019, more than 33 million cases of RSV-related lower respiratory tract infection (LRTI) occurred among children under 5 years of age globally, with more than 3.6 million hospitalizations and over 100,000 deaths (Li et al, 2022b). RSV poses serious health threats to children and elderly individuals worldwide, imposing a substantial disease and healthcare burden, particularly on governments in developing countries.

RSV is an enveloped, negative-strand RNA viruses, belonging to the *Pneumoviridae* family and the *Orthopneumovirus* genus (Rima et al, 2017). The RSV genome is 15.2 kb in length and contains 10 genes, encoding 11 proteins (Battles and McLellan, 2019). The RSV fusion (F) and attachment (G) glycoproteins are the main glycoproteins on the surface of the virus and play important roles in mediating virus entry into cells (Collins et al, 2013). RSV is classified into two antigenic subtypes (RSV A and RSV B) on the basis of the variability of G protein. In contrast, the F protein is highly conserved, with a sequence divergence of only 10% (Detalle et al, 2016; Ogunsemowo et al, 2018). The F protein is a type I transmembrane glycoprotein that exists as a trimer and consists of two conformational forms, prefusion (Pre-F) and postfusion (Post-F) (Anderson et al, 1992; Sugrue et al, 2001). During the process of RSV fusion with the host cell membrane, the F protein undergoes a conformational transition from Pre-F to Post-F (Battles and McLellan, 2019). RSV F has been characterized as containing at least six different antigenic epitopes, namely site Ø, I, II, III, IV, and V epitopes (Huang et al, 2019). Among them, sites II and III, and IV are present in both the Pre-F and Post-F conformations, whereas sites Ø and V are exclusive to the Pre-F conformation and are sensitive to neutralizing antibodies. Notably, Stabilization of the metastable prefusion conformation of the RSV F protein was achieved by introducing a disulfide bond and two cavity-filling substitutions, generating the variant known as DS-Cav1 (McLellan et al, 2013a). Subsequent iterative optimization of the stability and immunogenicity of DS-Cav1 led to the successful development of the second-generation F protein, DS2 (Joyce et al, 2016). This advancement overcame a major barrier in RSV vaccine and

[1]Department of Infectious Disease, The First Affiliated Hospital of USTC, Division of Life Sciences and Medicine, University of Science and Technology of China, Hefei, Anhui 230001, China. [2]Division of Life Sciences and Medicine, University of Science and Technology of China, Hefei, Anhui 230026, China. [3]Key Laboratory of Anhui Province for Emerging and Reemerging Infectious Diseases, Hefei, Anhui 230026, China. [4]CAS Key Laboratory of Special Pathogens and Biosafety, Wuhan Institute of Virology, Chinese Academy of Sciences, Wuhan 430071, China. [5]These authors contributed equally: Qianqian Wang, Xianliang Ke, Entao Li. ✉E-mail: shubo@wh.iov.cn; qiux@ustc.edu.cn

antibody-based therapeutic development, firmly establishing RSV Pre-F as a principal target for both vaccine design and antibody-derived therapies.

To date, the only RSV antibodies approved for marketing are palivizumab and nirsevimab, both of which are indicated for prophylactic use. Palivizumab is restricted to high-risk infants due to its monthly dosing regimen and elevated cost, limiting its clinical application and broader utilization (American Academy of Pediatrics Committee on Infectious & American Academy of Pediatrics Bronchiolitis Guidelines, 2014; Packnett et al, 2022). In contrast, nirsevimab is a fully human monoclonal antibody that is optimally engineered from the monoclonal antibody D25, which targets the Pre-F protein conformational epitope Ø and effectively neutralizes both the RSV A and RSV B strains (Zhu et al, 2017). Nirsevimab provides rapid protection against RSV in neonates and infants via a single-dose regimen to prevent lower respiratory tract disease caused by RSV. However, nirsevimab should not be administered to patients with active RSV infection, as no evidence exists for its therapeutic efficacy against established disease. Therefore, the development of therapeutic RSV antibody drugs is an urgent medical need.

Camelids produce a type of heavy chain-only antibodies (HCAbs), which is composed of the CH2 and CH3 constant regions, the hinge region, and the variable domain (VHH) (Greenberg et al, 1995; Mitchell and Colwell, 2018). VHH, which is commonly referred to as a nanobody, retains full antigen binding ability and has a molecular weight of only 15 kDa, approximately one-tenth that of conventional antibodies. Nanobodies (Nbs) have distinct advantages over conventional antibodies: (1) The extended CDR3 loop enables targeting of cryptic epitopes; (2) Exceptional physicochemical stability and solubility; (3) Ease of engineering; (4) Efficient expression in prokaryotic systems (e.g., *E. coli*) with high yields and reduced production costs; (5) High sequence homology between alpaca-derived Nbs and human VH3 family domains, enabling straightforward humanization and low immunogenicity in humans (Muyldermans, 2013).

In this study, dromedary camel was immunized with a mixture of RSV F protein antigens (DS-Cav1 and DS2) and a nanobody phage library was established. Multiple high-affinity, neutralizing Nbs were isolated from the library via phage display technology. Through rigorous enzyme-linked immunosorbent assay (ELISA) and microneutralization assays, Nbs 1G9 and 1D8 demonstrated potent binding affinity and neutralization capacity against RSV. Critically, 1G9 exhibited striking prophylactic and therapeutic efficacy in a murine model-addressing a significant gap in current RSV therapeutics. These results establish a robust preclinical foundation for developing novel RSV antibody therapeutics.

# Results

## Isolation and characterization of anti-RSV F Nbs

To obtain Nbs that specifically bind to the RSV F protein, two forms of RSV Pre-F proteins, DS-Cav1 and DS2, were produced via a eukaryotic expression system in HEK 293F cells. Reduced SDS-PAGE and Western blot analysis revealed that both antigenic proteins had a single band of the expected size and a high degree of protein purity (Appendix Fig. S1A,B). A dromedary camel was immunized with a mixture of the

two antigens DS-Cav1 and DS2 proteins at three-week intervals for a total of four times, and serum was collected to determine the binding titers for RSV Pre-F (DS-Cav1 and DS2). Clear seroconversion was observed after immunization, as shown by high end-point binding titers compared with pre-immunization serum (Appendix Fig. S1C,D). Then, peripheral blood mononuclear cells (PBMCs) were isolated from blood, performed RNA extraction, reverse transcription and obtained cDNA library. VHH coding regions were amplified, cloned into a phagemid vector and established a VHH phage-display library with about $3.7 \times 10^7$ independent clones (Appendix Fig. S2A,B). After two rounds of panning targeting the antigenic proteins DS-Cav1 and DS2, we ultimately obtained two libraries with capacities of $5.6 \times 10^5$ and $3.6 \times 10^5$, respectively (Appendix Fig. S2C). Individual phages were randomly picked, and their RSV F-binding activity was evaluated with phage ELISA. The heatmap showed that 95 and 94 out of the wells analyzed from two 96-well plates were positive for DS-Cav1 and DS2 binding, respectively, further confirming the enrichment of RSV F-binding phages, the positive rate is over 95% (Fig. 1A,B). On the basis of sequence diversity (representative sequences from each cluster) and binding affinity, we expressed and purified 55 candidate nanobodies. Through a high-throughput RSV neutralization assay, we identified several candidate antibodies exhibiting favorable neutralization activity (Appendix Figs. S3 and S4). Taking antibody yield into consideration, we ultimately selected the nanobodies 1G9 and 1D8 for further study.

To further understand the biological properties of the candidate antibodies, VHH coding regions were cloned into a human IgG1 Fc expression vector, which generates bivalent Fc-fusion nanobodies with extended half-life and effector functions, or into a His-tag expression vector for other in vitro assays (Fig. 1C) and were expressed in HEK293F cells. SDS–PAGE analysis showed that the nanobodies 1G9 and 1D8 had bands of anticipated molecular weight, with negligible impurities, and could be used for further research (Fig. 1D,E).

## The Nbs 1G9 and 1D8 prefer to bind to both the prefusion F variants

To characterize the binding activity of the purified candidate Nbs, we carried out ELISA assays using different RSV F proteins (DS-Cav1, DS2, and Post-F) as antigens. The results confirmed that Nbs 1G9 and 1D8 preferred to bind to both the Pre-F variants, DS-Cav1 and DS2, rather than Post-F. The 50% effective concentration ($EC_{50}$) values of their binding to Pre-F proteins ranged from 0.2 nM to 0.245 nM for 1G9-Fc (Fig. 2A) and 0.139 nM to 0.213 nM for 1D8-Fc (Fig. 2B). However, the binding activity of the antibody candidate is not comparable to that of the commercially available palizumab and nirsevimab (Fig. 2C,D). Importantly, the characteristic of binding activity does not entirely determine whether it can effectively neutralize the RSV subtypes. Additionally, we used strep-tagged DS2 as the antigen to determine the binding activity of Nbs 1G9-His and 1D8-His. The results showed that the $EC_{50}$ values for the binding activity of 1G9-His and 1D8-His were 0.1787 nM and 0.1964 nM, respectively (Fig. 2E). To rule out non-specific binding, we selected the phylogenetically related hMPV F protein and five distantly related viral envelope glycoproteins. The results showed that Nbs 1G9-Fc and 1D8-Fc specifically bound to RSV F protein, confirming the high specificity of the candidate antibodies (Fig. 2F).

To further determine whether the Nbs 1G9-Fc and 1D8-Fc could bind to the native RSV F protein on the membrane, we

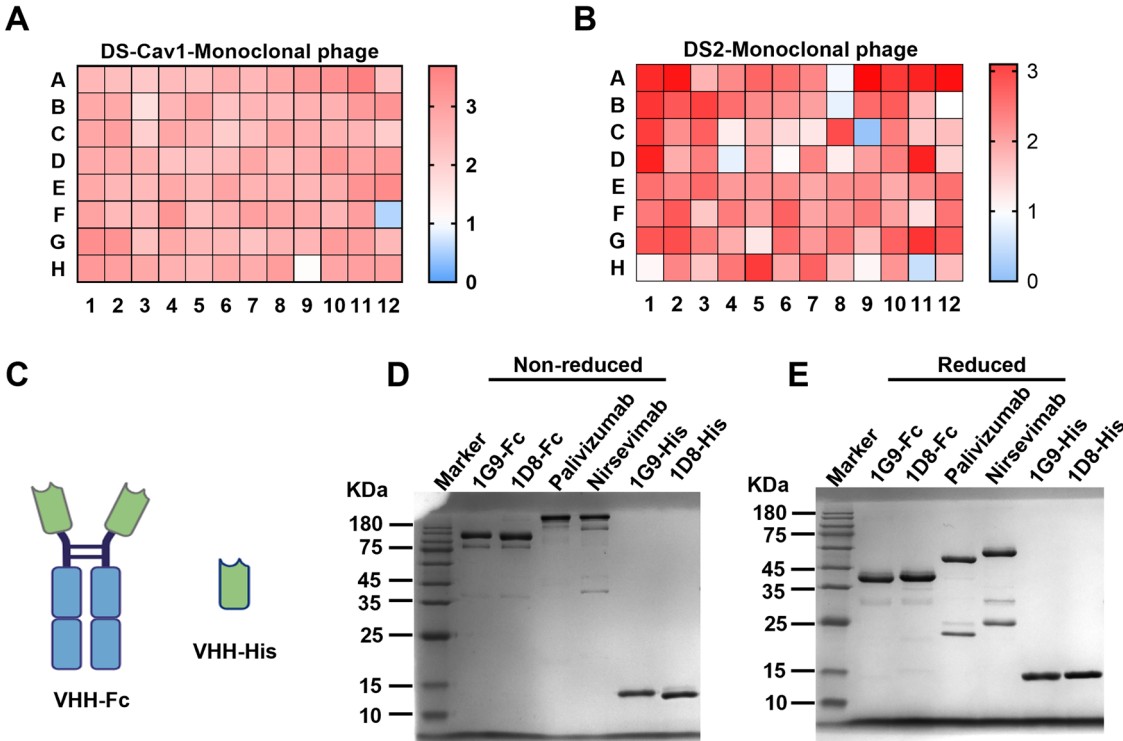

**Figure 1. Isolation and characterization of anti-RSV F Nbs from a phage display library.**

(A, B) Enrichment of monoclonal phages following two rounds of panning on RSV Pre-F (DS-Cav1 and DS2). Subsequently, the binding activity of the 96 isolated monoclonal phages was assessed by ELISA. Data are presented as a heatmap of $OD_{450}$ values, with the minimum value set at 0 and the baseline value at 1. (C) Schematic diagram of Nbs 1G9 and 1D8 fused to human IgG1 Fc or a His tag, respectively. (D, E) The Nbs 1G9 and 1D8 were expressed as fusions with the human IgG1 Fc or His tag in HEK293F cells. The resulting fusion proteins were evaluated by SDS-PAGE analysis with Coomassie blue staining. Palivizumab and nirsevimab, both monoclonal antibodies, were used as controls. Source data are available online for this figure.

performed an immunofluorescence assay. The results indicated that the fluorescence signals of the candidate Nbs were distinct and that they specifically recognized the native F protein (Appendix Fig. S5). For native RSV F protein, Nbs 1G9-Fc and 1D8-Fc exhibit comparable binding activity to other best-in-class RSV monoclonal antibodies.

## Neutralization of RSV subtypes by Nbs 1G9 and 1D8

To evaluate the neutralizing capacity of candidate Nbs 1G9 and 1D8 against RSV, we performed a neutralization immunofluorescence assay, a high-throughput screening method for assessing candidate Nbs. In brief, Nbs were serially diluted and incubated with RSV before being inoculated into Vero cells in 96-well plates. The maintenance medium was then changed, and the cells were fixed upon lesion formation. Finally, an immunofluorescence assay was performed using an RSV F protein-specific antibody to assess viral infection. The results showed that Nbs 1G9 and 1D8 were able to neutralize RSV A (strain A2), RSV B (strain CH93(18)-18) and RSV Long subtypes. Specifically, the 1G9-Fc fusion protein demonstrated potent neutralizing activity against RSV strains, with half-maximal inhibitory concentration ($IC_{50}$) values of 0.47 nM for RSV A2, 0.56 nM for RSV B, and 2.05 nM for RSV Long. The 1D8-Fc fusion protein exhibited neutralizing activity against RSV strains, with $IC_{50}$ values of 0.66 nM for RSV A2, 0.41 nM for RSV B, and 3.00 nM for RSV Long (Fig. 3A–C; Appendix Fig. S6).

Compared to palivizumab, 1G9 and 1D8 showed a lower $IC_{50}$ to neutralize RSV A2, RSV B and RSV Long and exhibited at least a 10-fold greater neutralization capacity against RSV A2 and RSV B. Although 1G9 and 1D8 exhibited lower neutralization capacity against RSV A2 and RSV Long strains compared to nirsevimab, they achieved complete neutralization of the RSV B strain, whereas nirsevimab failed to neutralize this strain.

Given that the candidate Nbs can bind to RSV Pre-F with high affinity, they effectively prevent the transition of the F protein from its prefusion conformation to its postfusion state, preventing the virus from entering the cell. Next, we further investigated whether the Nbs 1G9 and 1D8 could block the membrane fusion process after the viral particles have attached to the cells and prevented infection. We performed an RSV fusion inhibition assay in which RSV was incubated with cells at low temperature prior to attachment. A gradient dilution of the candidate nanobody was then added to the cells for incubation, and the cell lesions were fluorescently stained and quantified. The results demonstrated that both Nbs 1G9 and 1D8 effectively prevented infection when administered post-attachment of virions, with 1G9 exhibiting superior neutralization efficacy compared to 1D8 (Fig. 3D–F; Appendix Fig. S7). Similarly, 1G9 still exhibits superior membrane fusion inhibition activity against RSV B compared to nirsevimab. In summary, 1G9 exhibited potent and broad-spectrum neutralizing activity in vitro, inhibiting RSV infection by blocking post-attachment membrane fusion.

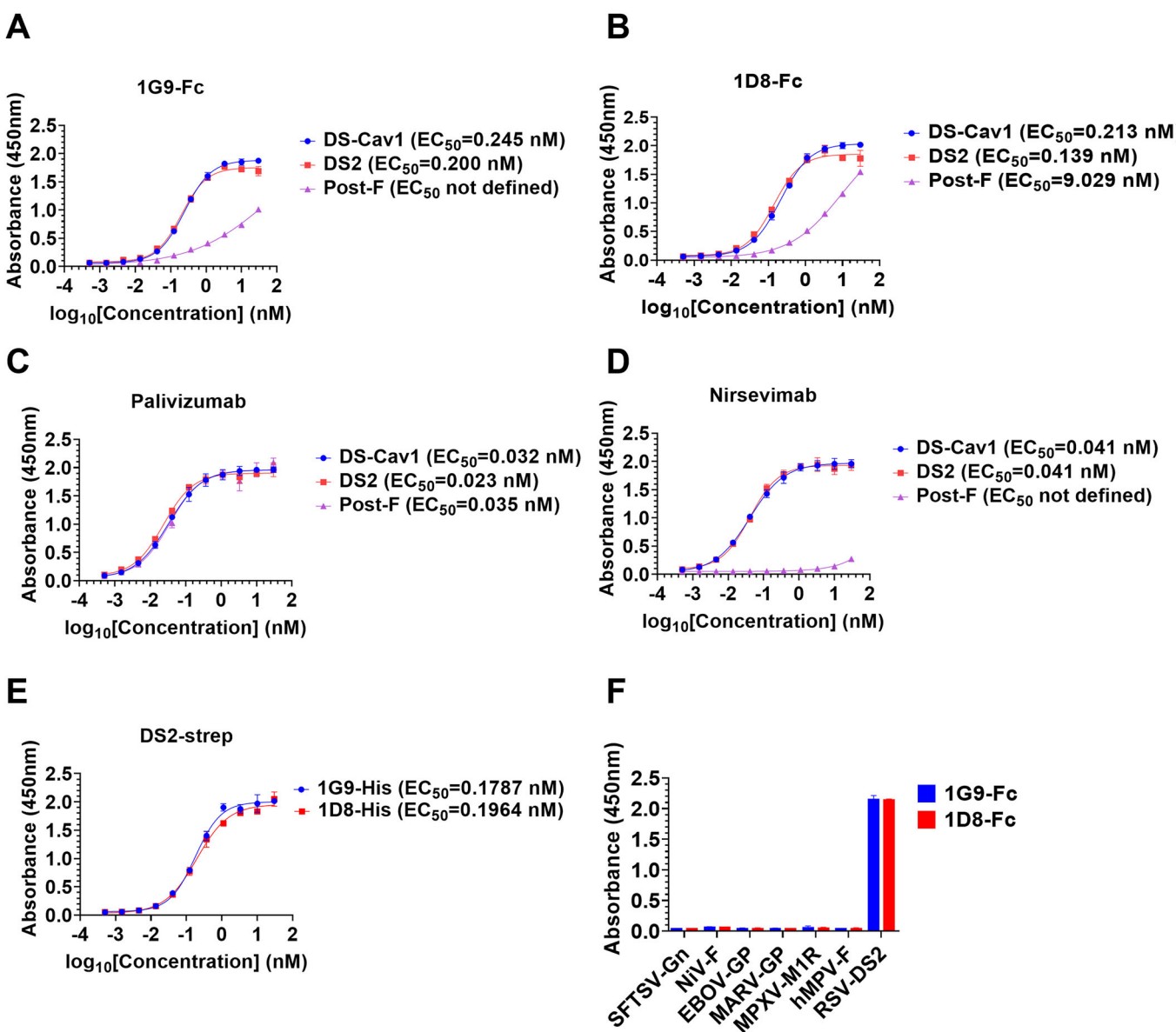

**Figure 2. Binding of the candidate Nbs to different F proteins.**

(A–D) The binding affinities of Nbs 1G9-Fc and 1D8-Fc to DS-Cav1, DS2 and Post-F were measured by ELISA ($n = 3$). Palivizumab and nirsevimab were used as controls. Dose–response curves were fitted to the $OD_{450}$ of the serially diluted Nbs to calculate the $EC_{50}$. Error bars indicate mean ± SD from three independent experiments. (E) Binding of Nbs 1G9-His and 1D8-His to DS2 (strep tag) was detected by ELISA ($n = 3$). (F) The binding capacity of 1G9-Fc and 1D8-Fc to surface glycoproteins of different viral strains by ELISA ($n = 2$). Data represent the $OD_{450}$ values measured at 30 nM, and are depicted as the mean ± SD from three independent experiments. Source data are available online for this figure.

## 1G9 and 1D8 bind a similar epitope on prefusion F

To gain insight into the antigenic binding sites that are targeted by Nbs 1G9 and 1D8, the cross-competition binding experiments were performed using RSV F protein. To date, at least six different RSV F binding sites, including Ø, I, II, III, IV, and V, have been identified (Huang et al, 2019). We subsequently used a set of monoclonal antibodies with known epitopes as competing antibodies, as follows: 131-2a (I) (Sominina et al, 1995), palivizumab (II) (Johnson et al, 1997), MPE8 (III) (Corti et al, 2013), 101F (IV) (McLellan et al, 2010), hRSV90 (V) (Mousa et al, 2017), and

nirsevimab (Ø) (Zhu et al, 2017), respectively. Nbs 1G9 and 1D8 were labeled with His-tag as a blocking antibody and used for competitive ELISA. The results showed that the Nbs 1G9 and 1D8 competed with each other for binding to immobilized RSV prefusion F and thus likely bound to overlapping epitopes (Fig. 4A,B). Specifically, only 101F was hindered by 1G9 and 1D8, indicating that they may bind to site IV. Overall, the results indicate that the Nbs 1G9 and 1D8 mediate broad neutralizing activity for diverse RSV A, RSV B and RSV Long strains by binding to similar antigenic sites (Fig. 4C).

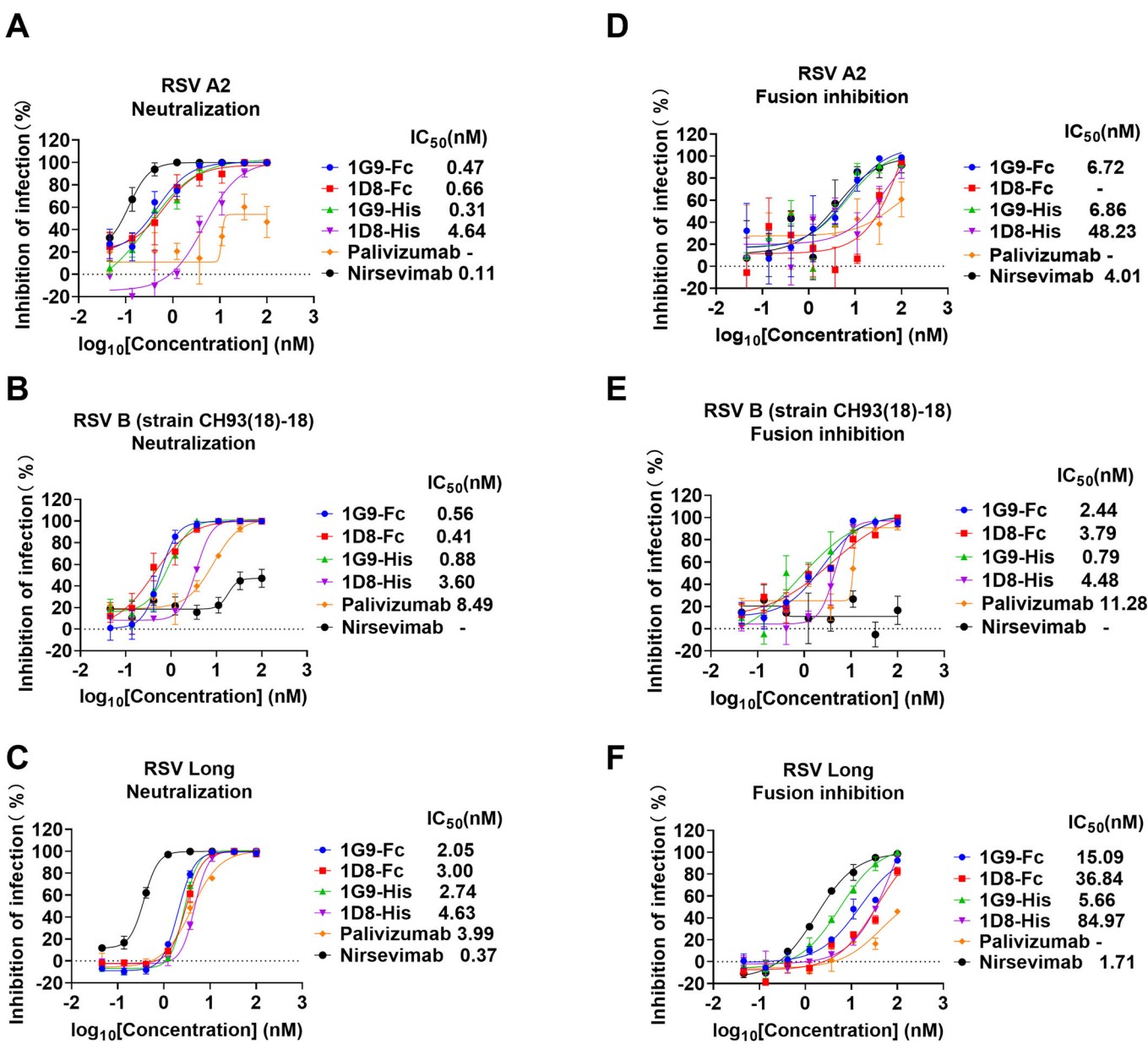

**Figure 3. Neutralization and fusion inhibition of RSV subtypes by Nbs 1G9 and 1D8.**

(A–C) Neutralization curves for Nbs 1G9, 1D8 and control antibody (palivizumab and nirsevimab) against RSV A2, RSV B (strain CH93(18)-18) and RSV Long ($n = 3$). (D–F) Membrane fusion inhibition curves for Nbs 1G9, 1D8 and control antibody (palivizumab and nirsevimab) against RSV A2, RSV B (strain CH93(18)-18) and RSV Long ($n = 3$). IC$_{50}$ values of Nbs 1G9, 1D8 and control antibody (palivizumab and nirsevimab) were determined by indirect immunofluorescence assay. The IC$_{50}$ values were calculated using GraphPad Prism 8.0 software. The means ± SEM from three independent experiment is depicted for RSV A, RSV B and RSV Long strains. Source data are available online for this figure.

## 1D8 and 1G9 target a defined structural pivot site on RSV F protein

To elucidate the molecular basis of RSV neutralization by the two Nbs, we determined the structures of Nbs complexed with the Pre-F protein (DS-Cav1) by cryo-electron microscopy (cryo-EM). The resolutions obtained for the 1G9 and 1D8 complexes with Pre-F were 2.77 Å and 3.26 Å, respectively (Appendix Table S1; Fig. 5A–D; Appendix Figs. S8 and S9).

Cryo-EM structures revealed that each nanobody (1D8 or 1G9) binds to a single protomer of the Pre-F trimer in a one-to-one manner, resulting in a symmetric arrangement of three Nbs around the trimer's threefold axis (Fig. 5A–D). Specifically, 1D8 and 1G9 recognize the "waist" region of the Pre-F monomer, corresponding to antigenic site IV, consistent with competition binding results using antibody 101 F. On each F protomer, 1D8 and 1G9 bury surface areas of 827 Å² and 808 Å², respectively. Notably, the two Nbs exhibit a highly overlapping binding interface. They share 20

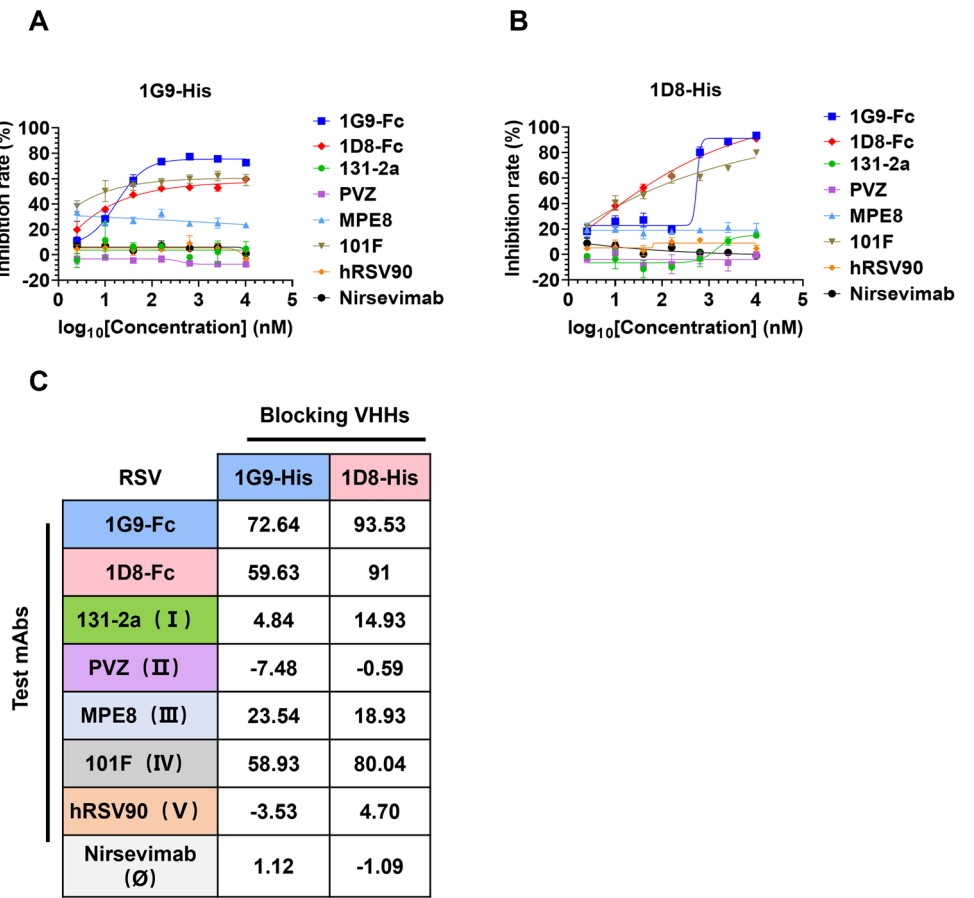

**Figure 4. 1G9 and 1D8 bind a similar epitope on Pre-F.**

(A, B) The competitive binding of Nbs 1G9 and 1D8 with six known epitope-specific antibodies. The values are normalized as competitive rates related to the negative control. Data represent the $OD_{450}$ values measured and are depicted as the mean ± SD from three independent experiments. (C) The values in squares are the percent binding of the monoclonal antibody (IgG) in the presence of the competing Nbs (His-tag) relative to a mock-competition control. Source data are available online for this figure.

interface residues, accounting for 95.2% (20/21) of the 1G9 epitope and 90% (20/23) of the 1D8 epitope (Appendix Fig. S11A). The convergence of both Nbs on the same epitope, combined with their potent neutralizing activity, suggests that this site is functionally critical and represents an immunodominant region on the F protein.

To elucidate the functional relevance of this epitope, we analyzed its structural characteristics. Both 1G9 and 1D8 are capable of binding to the Pre-F and Post-F conformations, indicating that the epitope resides within regions conserved across both states. As expected, structural alignment of the Pre-F and Post-F conformations (PDB: 3RKI) revealed a conformationally stable segment within site IV, spanning residues 411–458 of domain II on F1 (Fig. 5E), which is preserved in both states. This stable region is followed by a metastable β-strand (residues 466–468) located within the flexible HRB helix, which shifts toward the N-terminus of F2 to form the six-helix bundle during the transition from pre- to post-fusion.

Surprisingly, both 1G9 and 1D8 not only engage the structurally stable region of site IV—explaining their ability to bind both conformations—but also interact with the adjacent flexible segment

(Fig. 5F,G). While the epitope serves as a pivot point for structural transition, the Nbs act as molecular clamps, stabilizing β466–468 by anchoring it to the rigid portion of site IV and thereby preventing its unfolding. Furthermore, the localization of their primary epitopes within the structurally stable region likely accounts for their capacity to bind the Post-F conformation.

## Structural basis for broad neutralization and epitope conservation

To elucidate the mechanisms underlying the potent neutralization of both RSV A2 and B strains by these Nbs, we analyzed their interactions with the RSV F protein in detail. The paratope of nanobody 1G9 forms eleven hydrogen bonds with seven amino acid residues within antigenic site IV of the F protein (Fig. 6A). Specifically, the CDR2 region of 1G9 (residues Y62, S59, and D57) forms four hydrogen bonds with two positively charged residues, R429 and K433, located on the β430–435 strand. Additionally, the CDR3 region (residues L107 and W105) interacts with the β438–445 strand (residues S443 and K445) in the stable region through three hydrogen bonds. Extensive interactions also occur

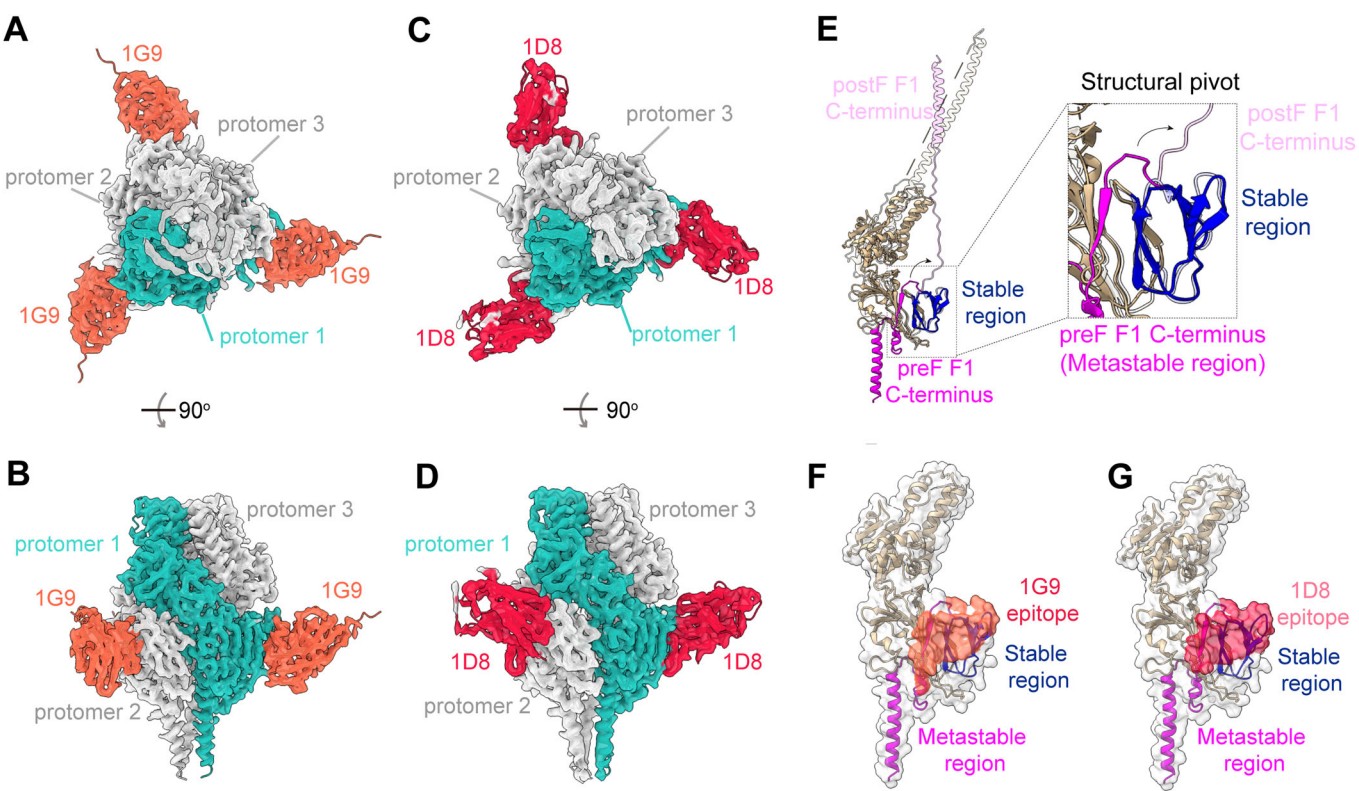

**Figure 5.  Cryo-EM structures of 1G9 and 1D8 bound to a structural pivot site on RSV A2 Pre-F.**

(A–D) Top and side views of Nbs 1G9 (A, B) and 1D8 (C, D) bound to the RSV A2 DS-Cav1 protein. Nbs 1G9 (shown in tomato) and 1D8 (shown in crimson) bind to protomers of the F protein, which are colored light sea green (protomer 1) and light gray (protomer 2 and protomer 3). (E) Superposition of Pre-Fusion (Pre-F) and post-fusion (Post-F) atomic models reveals a structural pivot site on antigenic site IV. The prefusion (Pre-F) and postfusion (Post-F, PDB: 3RKI) structures were aligned based on the stable region (residues 410–458, colored blue in Pre-F and light blue in Post-F) using ChimeraX. During the transition from Pre-F to Post-F, a metastable region following the stable region which links the C-terminal region of F1, extends toward the fusion loop (indicated by arrow), highlighted in magenta in Pre-F and light magenta in Post-F. The pivot site, consisting of both the stable and metastable regions, is boxed and shown in an enlarged view. (F) The 1G9 binding epitope and (G) the 1D8 binding epitope are mapped onto the structural pivot site of Pre-F. The Pre-F protomer is shown as a ribbon, superimposed with its surface. The 1G9 or 1D8 binding epitopes on the surface are colored tomato or crimson, respectively, while the rest of the Pre-F surface is colored white. The stable and metastable regions are shown as blue and magenta ribbons, respectively. 1G9 and 1D8 are omitted for clarity. Source data are available online for this figure.

between the CDR2 and CDR3 domains and the metastable β465–469 strand. In particular, three hydrogen bonds are established between CDR3 residues N102, A103, and A114 and the β466–468 segment (residues K465, S466, and Y468) (Fig. 6A). An additional hydrogen bond between CDR2 residue S56 and residue K470 further reinforces the interaction with the metastable region.

Similarly, nanobody 1D8 recognizes an epitope within antigenic site IV that closely resembles that of 1G9 (Fig. 6B). In this case, the FR3 domain forms a hydrogen bond between residues N74, N77, and R429, which lies in the loop connecting β438–443 and β449–453, forming the stable region of antigenic site IV. Furthermore, G102 on CDR3 forms a hydrogen bond with S443, stabilizing the interaction with the rigid region. On the metastable region, six hydrogen bonds are formed between residues K465, S466, Y468, and K470 and residues D105, T106 and C108 of the CDR3 domain (Fig. 6B).

We further compared the binding epitopes of 1G9 and 1D8 with those of previously reported RSV F antibodies, including hRSV90 (sites Ø, II, V) (Mousa et al, 2017), AM14 (sites IV, V) (Harshbarger et al, 2021), nirsevimab (site Ø) (Tang et al, 2019),

motavizumab (site II) (Gilman et al, 2015), MPE8 (sites II, III, IV, V) (Wen et al, 2017), and RB1 (site IV) (Tang et al, 2019) (Appendix Fig. S10). Among these, hRSV90, Motavizumab, and nirsevimab recognize epitopes that are largely distinct from those of 1G9 and 1D8, with minimal overlap. MPE8 and AM14 show partial overlap, but their binding spans multiple antigenic sites and only partially covers site IV. AM14 partially engages the pivot region but, as a trimer-specific antibody, binds across two protomers and cannot bind the postfusion conformation, suggesting that the stable region of the pivot contributes minimally to its binding (Appendix Fig. S11B–E).

In contrast, RB1 exhibits the highest degree of epitope similarity to 1G9 and 1D8 (Fig. 6C,D). Like the Nbs, RB1 targets the conserved site IV and binds both the Pre-F and Post-F conformations via its VH domain. However, the RB1 binding residues (defined by a 3.5 Å distance cutoff) (Tang et al, 2019) or hydrogen bond-forming residues (R429, K433, D440, S443, and V446) are confined to the stable region of the pivot. In addition, RB1's VL domain extends toward site V on an adjacent protomer—a region with lower sequence conservation and higher structural variability. In contrast, 1G9 and 1D8 adopt a more focused and compact

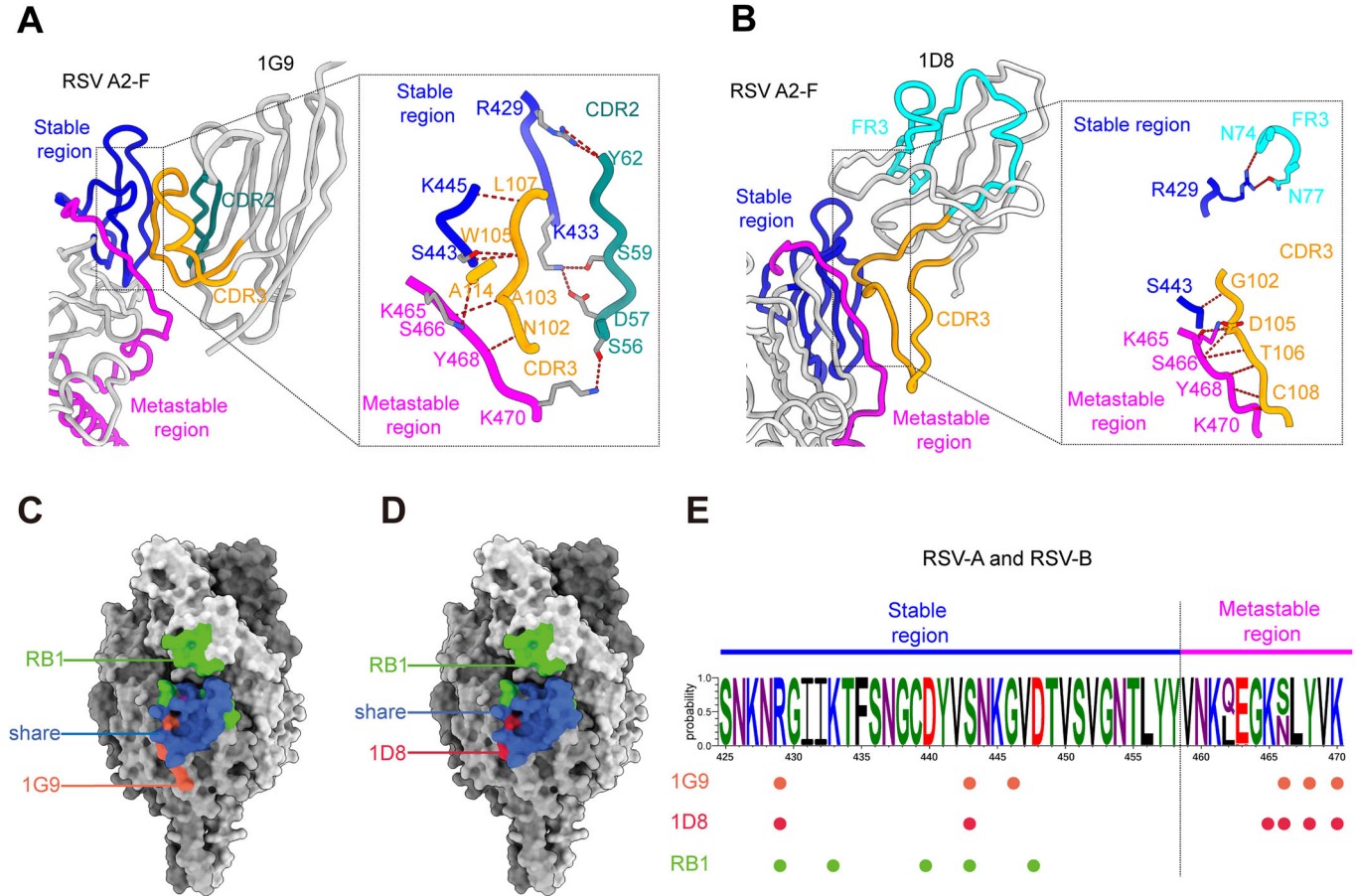

**Figure 6.  Recognition sites of the Nbs on the RSV-F protein.**

(A) Interaction details of 1G9 with RSV F. The stable and metastable regions of the structural pivot site, along with interaction residues, are colored blue and magenta, respectively. The CDR2 and CDR3 loops of 1G9 are colored light sea green and orange. The interface is boxed and shown in an enlarged view. Hydrogen bonds between 1G9 and the F protein are indicated by red dashed lines. (B) Interaction details of 1D8 with RSV F. The stable and metastable regions of the structural pivot site, along with interaction residues, are colored blue and magenta, respectively. The CDR1 and CDR3 loops of 1D8 are colored cyan and orange. The interface is boxed and shown in an enlarged view. Hydrogen bonds between 1D8 and the F protein are indicated by red dashed lines. (C, D) Comparison of the interfacing areas of 1G9 (C) and 1D8 (D) with the monoclonal antibody RB1 on DS-Cav1. The RB1 interacting residues (identified by PDBePISA) are colored green, the interfacing areas of 1G9 and 1D8 are colored orange and red, respectively, and the residues shared by RB1 with 1G9 or 1D8 are highlighted in blue. (E) Sequence conservation analysis of the 1G9 and 1D8 interacting residues on RSV-F protein. The sequences of the RSV F protein (subtype A: 5878 sequences; subtype B: 4799 sequences) obtained from the NCBI database were aligned using MAFFT, and residue frequencies were analyzed using SeqLogo2. Residues located in the stable and metastable regions of the structural pivot are separated by a dashed line. Residues contributing to strong interactions (hydrogen bonds or π–cation interactions) with 1G9, 1D8, or RB1 are highlighted as orange, crimson, or blue, respectively. Source data are available online for this figure.

binding mode entirely restricted to the conserved core of site IV. Shared residues including R429 and S443 in the stable region, as well as S466, Y468, and K470 in the metastable region, are engaged by both Nbs through hydrogen bonding, underscoring their role as structural clamps anchoring the conformational pivot (Fig. 6E).

To assess the evolutionary conservation of the nanobody binding interfaces, we retrieved full-length RSV F sequences from the NCBI database (5535 RSV A and 4343 RSV B strains), performed sequence alignments, and calculated conservation rates. The epitope region spanning residues 426–470—recognized by both 1D8 and 1G9—exhibited a high degree of conservation: over 99.95% in both RSV A and RSV B, except for residue 466 which is serine (S) in RSV A (98.93% conservation) and asparagine (N) in RSV B (99.86% conservation). Residues involved in hydrogen bonding or π–cation interactions with the Nbs also showed high

conservation, explaining their broad neutralization efficacy against both RSV subtypes (Appendix Table S2; Appendix Fig. S12).

## Prophylactic and therapeutic efficacy of candidate Nbs against RSV infection in mice

Next, the prophylactic efficacy and therapeutic efficacy of Nbs 1G9 and 1D8 in an animal model were evaluated (Fig. 7A). To prolong the in vivo serum half-life of the Nbs, each nanobody was cloned into the PTT5 vector with Fc region of human IgG1 and obtained the VHH-Fc fusion protein (1G9-Fc and 1D8-Fc). To compare the prophylactic efficacy of Nbs with nirsevimab, mice were intraperitoneally injected with Nbs or nirsevimab at a dose of 2 mg/kg one day prior to RSV A2 infection, while the control group received PBS. Twenty-four hours post-administration, all animals were

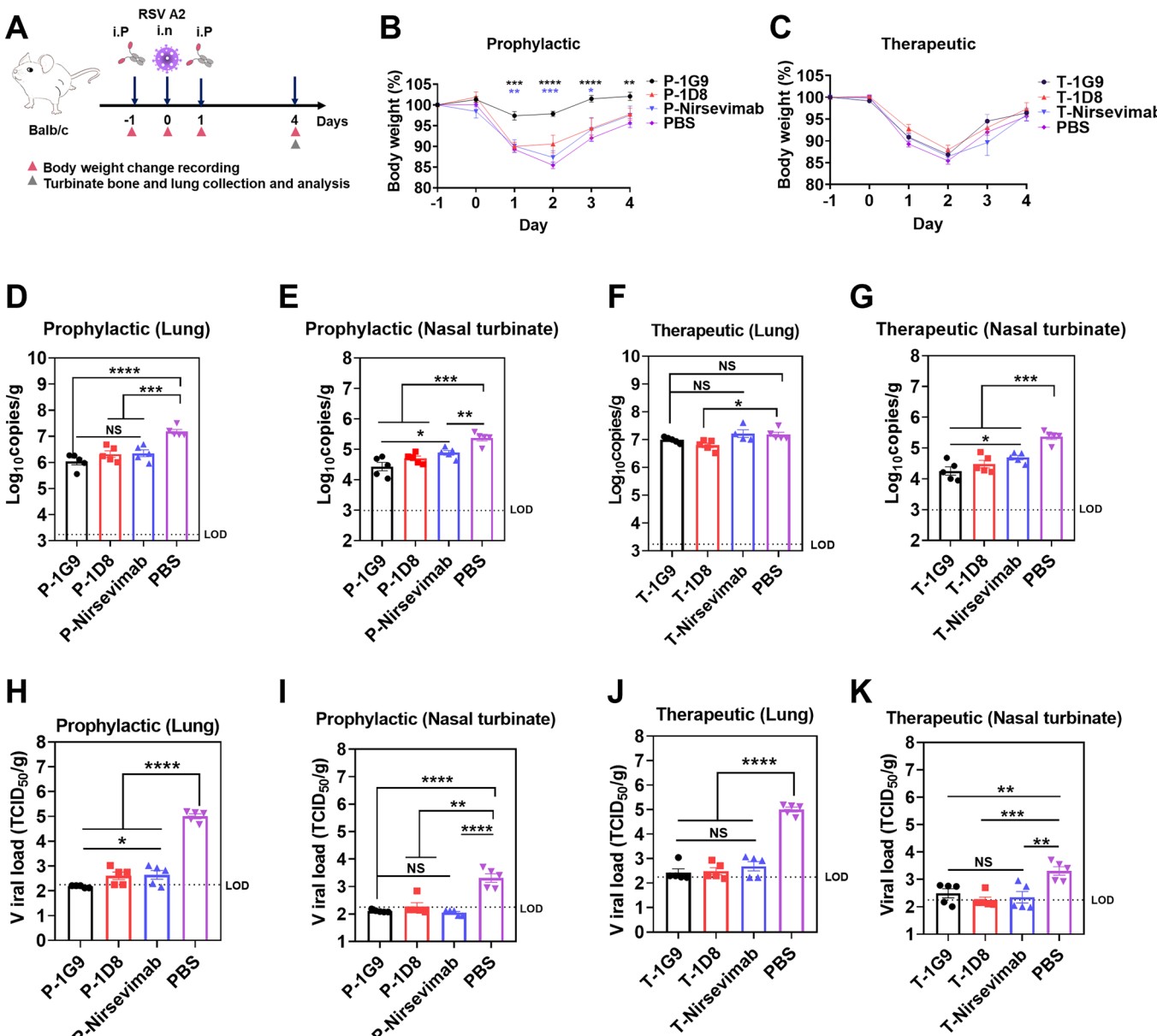

anesthetized and infected intranasally with RSV A2. Nasal turbinate and lung tissues were harvested on day 4 after challenge to assess the amount of viral RNA and virus titers. Compared with the PBS and nirsevimab groups, the nanobody 1G9 maintained body weight of the RSV-infected mice from day 1 postinfection in the prophylactic group (Fig. 7B). We found that the amount of viral RNA and titers in the nasal turbinate and lung tissues of our candidate nanobody-treated mice were also reduced (Fig. 7D,E and 7H,I).

Despite the significant challenges surrounding the feasibility of RSV neutralizing Nbs as RSV therapeutics, we have attempted to evaluate the potential of candidate Nbs as therapeutics in mouse models. Animals were administrated with high doses (8 mg/kg) of Nbs or nirsevimab 24 h after inoculation with RSV A2 (Fig. 7A). In the therapeutic model, RT‒qPCR showed a reduction in the amount of viral RNA in the nasal turbinate and lung tissues of

candidate nanobody-treated mice compared with PBS group (Fig. 7F,G). Specifically, viral RNA levels in the 1G9 group showed ~1.55-fold and ~12.97-fold reductions in lung tissues and nasal turbinate, respectively; 1D8 group showed ~2.42-fold and ~7.75-fold reductions in lung tissues and nasal turbinate, respectively. RSV titer results showed that the viral loads were largely eliminated in the nanobody group (Fig. 7J,K), although no significant improvement in body weight was observed in the 1G9 and 1D8 groups post-infection (Fig. 7C).

Additionally, Pathological analysis of lung sections from RSV A2-infected mice revealed well-preserved bronchial, bronchiolar, and alveolar structures in the 1G9, 1D8, and nirsevimab groups (Fig. 8). Collectively, these findings demonstrate that Nbs 1G9 and 1D8 hold considerable promise for both prophylactic (low-dose) and therapeutic (high-dose) applications against RSV.

**Figure 7. Prophylactic and therapeutic efficacy of candidate Nbs against RSV infection in mice.**

(A) Schematic diagram of the RSV prophylactic or therapeutic animal experiments. Balb/c mice were intraperitoneally injected with Nbs or nirsevimab 24 h before (2 mg/mL) and after (8 mg/mL) the challenge. The control group received an injection of PBS. Lung and nasal turbinate tissues from each group of mice were harvested at day 4 post-infection. (B, C) The body weights of all mice were monitored before and after infection with the virus ($n = 5$). The asterisks represent significance analyses of different groups (black:1G9 and PBS; blue: 1G9 and nirsevimab). Day1: 1G9 vs. PBS, $p = 0.0002$; 1G9 vs. nirsevimab, $p = 0.0042$; Day2: 1G9 vs. PBS, $p < 0.0001$; 1G9 vs. nirsevimab, $p = 0.003$; Day3: 1G9 vs. PBS, $p < 0.0001$; 1G9 vs. nirsevimab, $p = 0.039$; Day4: 1G9 vs. PBS, $p = 0.0024$; 1G9 vs. nirsevimab, $p = 0.0962$. (D) The viral RNA copy number in the lung tissues in prophylactic groups were detected by RT–qPCR ($n = 5$). P-1G9 vs. PBS, $p < 0.0001$; P-1D8 vs. PBS, $p = 0.0005$; P-nirsevimab vs. PBS, $p = 0.0007$; P-1G9 vs. P-nirsevimab, $p = 0.1287$. (E) The viral RNA copy number in the nasal turbinate tissues in prophylactic groups were detected by RT–qPCR ($n = 5$). P-1G9 vs. PBS, $p = 0.0005$; P-1D8 vs. PBS, $p = 0.0005$; P-nirsevimab vs. PBS, $p = 0.0038$; P-1G9 vs. P-nirsevimab, $p = 0.0212$. (F) The viral RNA copy number in the lung tissues in therapeutic groups were detected by qRT–PCR ($n = 5$). T-1G9 vs. PBS, $p = 0.0820$; T-1D8 vs. PBS, $p = 0.0123$; T-nirsevimab vs. PBS, $p = 0.8316$; T-1G9 vs. T-nirsevimab, $p = 0.1290$. (G) The viral RNA copy number in the nasal turbinate tissues in therapeutic groups were detected by qRT–PCR ($n = 5$). T-1G9 vs. PBS, $p = 0.0001$; T-1D8 vs. PBS, $p = 0.0004$; T-nirsevimab vs. PBS, $p = 0.0004$; T-1G9 vs. T-nirsevimab, $p = 0.0222$. (H) Viral titers in lung tissues in prophylactic groups were measured by RSV indirect immunofluorescence assay ($n = 5$). P-1G9 vs. PBS, $< 0.0001$; P-1D8 vs. PBS, $< 0.0001$; P-nirsevimab vs. PBS, $< 0.0001$; P-1G9 vs. P-nirsevimab, $p = 0.0287$. (I) Viral titers in nasal turbinate tissues in prophylactic groups were measured by RSV indirect immunofluorescence assay ($n = 5$). P-1G9 vs. PBS, $p < 0.0001$; P-1D8 vs. PBS, $p = 0.0012$; P-nirsevimab vs. PBS, $p < 0.0001$; P-1G9 vs. P-nirsevimab, $p = 0.3095$. (J) Viral titers in lung tissues in therapeutic groups were measured by RSV indirect immunofluorescence assay ($n = 5$). P-1G9 vs. PBS, $p < 0.0001$; P-1D8 vs. PBS, $p < 0.0001$; P-nirsevimab vs. PBS, $p < 0.0001$; P-1G9 vs. P-nirsevimab, $p = 0.2912$. (K) Viral titers in nasal turbinate tissues in therapeutic groups were measured by RSV indirect immunofluorescence assay ($n = 5$). P-1G9 vs. PBS, $p = 0.0075$; P-1D8 vs. PBS, $p = 0.0006$; P-nirsevimab vs. PBS, $p = 0.0063$; P-1G9 vs. P-nirsevimab, $p = 0.6032$. Data are presented as mean ± SEM, statistical analysis was performed using t tests (and nonparametric tests). The asterisks indicate significance: *$P < 0.05$, **$P < 0.01$, ***$P < 0.001$, ****$P < 0.0001$. No significance (NS). Source data are available online for this figure.

# Discussion

Respiratory syncytial virus (RSV) is the most common cause of acute respiratory infection in infants and young children (Li et al, 2022b; Openshaw et al, 2017; Shi et al, 2017). Currently, the commercially available RSV antibody drugs used for prevention include palivizumab and nirsevimab. Palivizumab, the first RSV monoclonal antibody marketed in 1998, is a humanized murine monoclonal antibody. The disadvantages of palivizumab include its high dose, frequent administration and high price, which limit its widespread use (IMpact-RSV Study Group, 1998; Johnson et al, 1997; Subramanian et al, 1998). Nirsevimab is the latest RSV antibody drug to reach the market in 2023 (Zhu et al, 2017). The antibody targets the Ø epitope of the Pre-F protein and is the world's first antibody drug used to prevent RSV infection in infants and young children for an entire season with a single dose (Fullarton et al, 2023; Muller et al, 2023). Both are used to prevent RSV infection in infants and young children. However, whether palivizumab and nirsevimab can effectively treat RSV infection is unknown, and the presence of escape mutants has been demonstrated in the clinical trial (Bin et al, 2019; Griffin et al, 2020; Sun et al, 2022). In our study, the nanobody 1G9 demonstrated potent neutralizing activity against RSV A2, RSV B (strain CH93(18)-18), and RSV Long strains, exhibiting a significantly greater neutralization effect than nirsevimab against RSV B. Importantly, nirsevimab was shown to neutralize the RSV B laboratory strain, with an $IC_{50}$ against RSV B9320 of approximately 3.1 ng/mL (Zhu et al, 2017). To gain insight into the mechanism of viral escape, we performed genetic sequence analyses of RSV B9320 and RSV B (strain CH93(18)-18), which showed that the RSV F gene presented four types of amino acid substitution at the nirsevimab-binding site Ø (Appendix Fig. S13). This further indicates that nirsevimab targets site Ø, which is associated with a risk of variation.

Moreover, in mouse model, the efficacy of 1G9-Fc (dose in mg/mL) in both prophylaxis and treatment of RSV is comparable to that of nirsevimab. Despite the effort for developing numerous RSV-neutralizing Nbs, such as m35 (His-tag) (Xun et al, 2021), F-VHH-4 (His-tag) (Rossey et al, 2017a) and ALX-0171 (trimeric

nanobody) (Detalle et al, 2016), none have yet been approved for clinical use. The 1G9-Fc nanobody developed in this study not only exhibits significantly enhanced neutralization potency against RSV B compared to nirsevimab but also holds promise for broader application due to its potential for greater site conservation and resistance profile. However, there are several limitations in this study. We performed in vivo animal protection tests against only the RSV A2. Although 1G9-Fc and 1D8-Fc have good neutralizing abilities against both RSV A and RSV B in vitro, testing the protective effects of antibodies against RSV B (strain CH93(18)-18) in vivo is important. On the other hand, owing to the limitations of the collection of different RSV clinical strains, it is currently not possible to perform a more comprehensive assessment of the broad spectrum of candidate Nbs. This is necessary for subsequent clinical trials for further development of neutralizing Nbs. In the therapeutic setting, we did not observe significant differences in body weight among groups. This is likely because a substantial change in viral burden is required to induce measurable weight loss in this mouse model, making body weight an insensitive readout for therapeutic assessment. Nevertheless, viral titration analysis demonstrated a clear reduction of viral load in nanobody-treated animals, indicating a robust in vivo antiviral effect despite the absence of detectable weight differences.

Structural studies of candidate Nbs bound to RSV F proteins reveal details of their interactions as well as their structural differences from known antibodies with previously identified epitopes. The results show that both 1G9 and 1D8 target a structure pivot site on antigenic site IV of RSV Pre-F, consisting of a structurally stable region shared by Pre-F and Post-F, as well as a metastable region, which links the C-terminus of F1 and unravels and rotates to form a 6-helix bundle for membrane fusion. Both 1G9 and 1D8 accurately encompass this pivot, crosslinking the stable part and the metastable helix by forming compact hydrogen bond networks with both parts, which may prevent structural changes during the transition from the Pre-F to Post-F. This specific binding stabilizes the Pre-F conformation of the virus, thereby effectively inhibiting viral membrane fusion. Notably, other antibodies, such as RB1 and AM14—both known for their broad

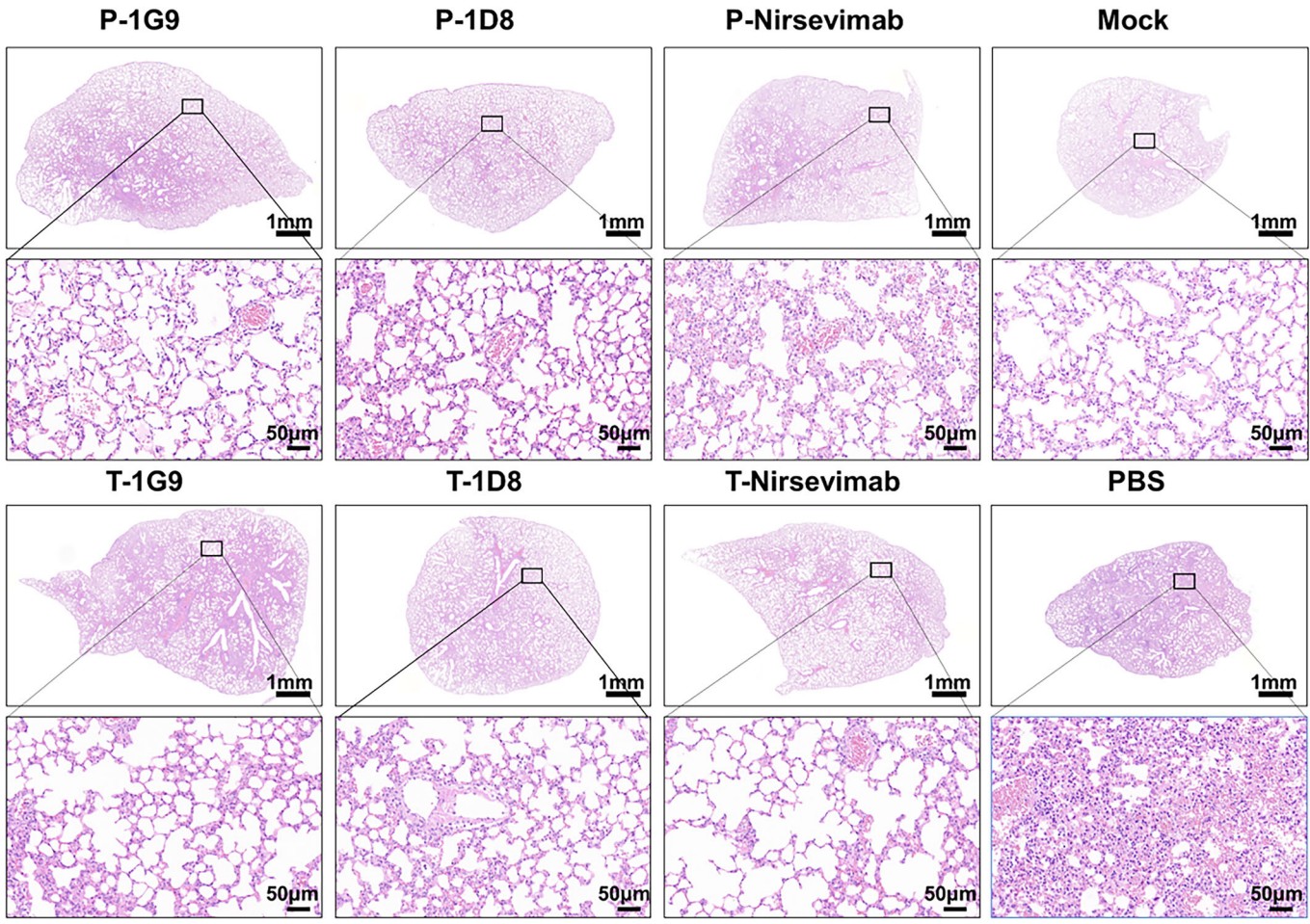

**Figure 8. Histopathological sections of mouse lung tissues.**

The prophylactic groups (P) included three mouse samples ($n = 3$), therapeutic groups (T) included two mouse samples ($n = 2$), and one representative image was shown. Mock is a blank control, and mice were not administered antibody and not inoculated intranasally with RSV. PBS is a negative control, and mice were intraperitoneally injected with PBS and infected with RSV A2. Source data are available online for this figure.

neutralization ability—share partial epitope overlap with 1G9 or 1D8. However, the binding residues of RB1, as well as those mediating strong interactions such as hydrogen bonds with the F protein, are confined exclusively to the stable region. There is no evidence that RB1 engages in strong interactions with the metastable region, suggesting that it may exert its neutralizing function through a mechanism distinct from that of 1G9 and 1D8. In contrast, the AM14 epitope spans multiple antigenic sites, including the pivot region—a structurally dynamic area. However, AM14 binds only to the Pre-F but not the Post-F, indicating that the stable region may play a limited role in its binding. This could be attributed to the relatively large binding interface typical of conventional antibodies and implies a more complex mode of action for AM14. In any case, the highly specific and selective binding of 1G9 and 1D8 to the pivot region highlights the functional importance of this epitope in viral replication. Site IV is the most conserved antigenic site in RSV-F. As a result, the 1G9 and 1D8 residues interfacing with RSV-F, including the residues involved in crosslinking by the Nbs, are highly conserved in all RSV strains. This explained the neutralization of the RSV A2 and RSV B

strains, and potentially broad neutralization activity against other RSV strains, although these effects have not been fully tested.

In summary, this study has structurally and functionally characterized 1G9 and 1D8 as potent broad-spectrum Nbs against RSV. These Nbs demonstrate unique molecular recognition of a conformational pivot site within antigenic site IV, effectively bridging the metastable HRB region with the structurally rigid domain II of the F protein. Comprehensive efficacy studies in animal models demonstrated that these Nbs effectively inhibit viral replication, exhibiting both preventive and therapeutic potential. The favorable pharmacological properties of these Nbs support their advancement to clinical trials for the prophylactic and therapeutic treatment against RSV infection in high-risk populations, particularly infants. Currently, 1G9 is engineered with incorporating Fc mutations (YTE) to extend serum half-life (Dall'Acqua et al, 2006; Dall'Acqua et al, 2002; Robbie et al, 2013), designed as passive immunoprophylaxis for RSV infection in infants. Furthermore, we have developed a 1G9-IgA fusion construct (VHH-IgA) (Li et al, 2022a) that represents an inhalable therapeutic modality. This cost-effective, minimally invasive

approach combining both prophylactic and therapeutic applications establishes a novel passive immunotherapy platform, offering new treatment strategies for RSV-associated diseases.

# Methods

### Reagents and tools table

| Reagent/Resource | Reference or Source | Identifier or Catalog Number |
| --- | --- | --- |
| **Experimental models** | | |
| BALB/cJGpt | GemPharmatech Co., Ltd (Nanjing, China) | N000020 |
| Vero | Baidi Biotech Ltd. | C5168 |
| HEK293F | Dr. Tengchuan Jin (the University of Science and Technology of China, Hefei, China) | NA |
| RSV A2 | Dr. Rui Gong (Wuhan Institute of Virology, Center for Biosafety Mega-Science, Chinese Academy of Sciences, Wuhan, Hubei, China) | 16533.06. IVCAS6.7566 [National Virus Resource Center (NVRC)] |
| RSV B [CH93(18)-18] | Dr. Rui Gong | 0810040CF (ZeptoMetrix) |
| RSV Long | Dr.Jiaming Lan (Shanghai Institute of Immunity and Infection, Chinese Academy of Sciences) | NA |
| **Recombinant DNA** | | |
| pLexm-DS-Cav1 | Dr. Rui Gong | NA |
| pLexm-DS2 | Dr. Rui Gong | NA |
| pLexm-Post F | Dr. Rui Gong | NA |
| PTT5 | Dr. Tengchuan Jin | NA |
| **Antibodies** | | |
| 131-2a | AntibodySystem | RVV02818 |
| MPE8 | AntibodySystem | RVV02814 |
| 101F | AntibodySystem | RVV02816 |
| hRSV90 | AntibodySystem | RVV02813 |
| Nirsevimab | AntibodySystem | DVV02802 |
| Palivizumab | Shanghai Universal Biotech Co., Ltd. | abs190405 |
| Rabbit anti-human IgG-Fc HRP | Sino Biological | 10702-T16-H-100 |
| HRP-conjugated 6*His, His-Tag Monoclonal antibody | Proteintech | HRP-66005 |
| FITC-conjugated anti-human IgG | Proteintech | SA00003-12 |
| **Chemicals, Enzymes and other reagents** | | |
| DMEM | ThermoFisher | C11995500BT |
| FBS | VivaCell | C04001-0500 |
| PS | Biosharp | BL505A |

| Reagent/Resource | Reference or Source | Identifier or Catalog Number |
| --- | --- | --- |
| Trypsin | Yeasen Biotechnology (Shanghai) Co., Ltd. | 40101ES |
| SMM293-TII-N | Sino Biological Inc. | RZ18JU-A |
| PBS | Servicebio | G4202 |
| Tissue fixation solution (4%PFA) | Servicebio | G1101 |
| Polyethylenimine (PEI) | MACKLIN | P924174-1g |
| 0.22 μm sterile syringe filters | Biosharp | BS-PES25-22-S |
| Amicon Ultra centrifugal filters | Millipore | UFC901096 |
| Tween 20 | Solarbio | T8221 |
| Tris | BioFroxx | 1115KG001 |
| imidazole | Aladdin | I108706-500g |
| NaCl | Sinopharm Chemical Reagent Co., Ltd. | 10019318 |
| Acetic acid | Sinopharm Chemical Reagent Co., Ltd. | 10000218 |
| Isopropanol | Sinopharm Chemical Reagent Co., Ltd. | 80109218 |
| Ethanol absolute | Sinopharm Chemical Reagent Co., Ltd | 10009218 |
| Skim Milk Powder | Biosharp | BS102-500g |
| sulfuric acid | Sinopharm Chemical Reagent Co., Ltd. | 110021618 |
| Glycerol | Biosharp | 56-81-5 |
| Agar | BioFroxx | 8211GR500 |
| Yeast extract | ThermoScientific | LP0021 |
| Tryptone | ThermoScientific | LP0042 |
| TMB substrate | Beyotime | P0209-500 mL |
| ELISA Plate | in vitro scientific | 100096H |
| PrimeSTAR Max DNA Polymerase R045Q | TAKARA | R045A |
| Virus DNA/RNA Extraction Kit 2.0 | Vazyme | RM401-04 |
| HiScript II One Step qRT–PCR SYBR Green Kit | Vazyme | Q221 |
| PrimeScript™ II 1st Strand cDNA Synthesis Kit | Takara | 6210 A |
| DNA Assembly Mix Mono | Yugong Biolabs, Inc. | EG25202-S/M |
| NI-NTA GBrose 6FF | GeneBio | GBP005-50 |
| E.Z.N.A.® Gel Extraction Kit | OMEGA | D2500-02 |
| E.Z.N.A.® Total RNA Kit I | OMEGA | IR6834 |
| D-Biotin (Vitamin H) | Solarbio | 58-85-5 |

| Reagent/Resource | Reference or Source | Identifier or Catalog Number |
|---|---|---|
| Protein A GBrose FF | GeneBio | GBP001-50 |
| STarm Beads 4FF | Smart-Lifesciences | SA092005 |
| FastPure Enhanced EndoFree Plasmid Maxi Plus Kit | Vazyme | DC222-01 |
| FastPure EndoFree Plasmid Mini Plus Kit | Vazyme | DC204-01 |
| **Other** | | |
| Microprotein purification system | Union-Biotech (Shanghai) Co., Ltd | UEV 25D |
| Electroporation System | Bio-Rad | Gene Pulser Xcell™ |
| ImageXpress MicroConfocal | Moleculardevices | 76177-140 |
| Microscope | OLYMPUS | CKX53 |
| Microplate Photometer | Thermo Scientific | Multiskan FC |
| NanoDrop | Thermo Scientific | 840-317400 |
| Applied Biosystems QuantStudio 3 | Thermo Scientific | A28137 |

## Cells, virus, and animals

Vero cells (African green monkey kidney epithelial cells, BDBIO, C5168) were grown in culture Dulbecco's modified Eagle medium (DMEM, ThermoFisher, C11995500BT) containing fresh 10% fetal bovine serum (FBS, VivaCell, C04001-0500), 1% penicillin–streptomycin solution (PS, Biosharp, BL505A). Adherent cells were cultured in a cell incubator with 5% $CO_2$ at 37 °C. HEK293F cells (human embryonic kidney cells) were kindly provided by Dr. Tengchuan Jin (the University of Science and Technology of China, Hefei, China), and were maintained at 37 °C with 5% $CO_2$ in an orbital shaker in medium (SMM293-TII-N, SinopecBiological, RZ18JU-A).

Respiratory syncytial virus strain A2 [A2; National Virus Resource Center (NVRC)] and RSV B [CH93(18)-18; ZeptoMetrix] were kindly provided by Dr. Rui Gong (Wuhan Institute of Virology, Center for Biosafety Mega-Science, Chinese Academy of Sciences, Wuhan, Hubei, China). RSV Long strain was from Dr. Jiaming Lan (Shanghai Institute of Immunity and Infection, Chinese Academy of Sciences). RSV strains were propagated and quantified in Vero cells by indirect immunofluorescence assay.

Eight-week-old, female Balb/c mice were purchased from the GemPharmatech Co., Ltd (Nanjing, China). Mice were housed in humidity and temperature-controlled, room with a 12:12 h light-dark cycle. The mice were capable of autonomously acquiring food and water. All procedures were in accordance with the guidelines of and approved by the University of Science and Technology of China (USTC) Animal Resources Center and University Animal Care and Use Committee (Permit Number: USTCACUC26100122093).

## Antigen expression and purification

The RSV F protein-encoding gene sequence and plasmids, including pLexm-DS-Cav1, pLexm-DS2, and pLexm-post F, were kindly provided by Dr. Rui Gong (Wuhan Institute of Virology, Center for Biosafety Mega-Science, Chinese Academy of Sciences, Wuhan, Hubei, China). The DNA sequences for RSV F were cloned into the mammalian expression vector PTT5, which contains a His6 tag, to construct recombinant plasmids: PTT5-DS-Cav1, PTT5-DS2, and PTT5-post F. Next, the target plasmids were transfected into suspension HEK293F cells at a density of 2.0 to $3.0 \times 10^6$ cells/mL by using polyethylenimine (PEI, MACKLIN, P924174-1g) following the manufacturer's instructions. The cell culture supernatant was harvested four days post transfection by centrifugation at 5000 rpm for 20 min at 4 °C. Then the supernatant was filtered with 0.22 μm sterile syringe filters (Biosharp; BS-PES25-22-S) and passed through the affinity column at a flow rate of 2.0 mL/s. The antigen was purified using a microprotein purification system (Union-Biotech (Shanghai) Co., Ltd; UEV 25D) with a 1 mL NI-NTA (GeneBio, GBP005-50) affinity column (binding buffer: 20 mM Tris, 500 mM NaCl, 20 mM imidazole, pH 8.0; elution buffer: 20 mM Tris, 500 mM NaCl, 400 mM imidazole, pH 8.0). The eluent corresponding to the absorption peak was collected and identified by reduced SDS–PAGE. Finally, the purified protein was concentrated with Amicon Ultra centrifugal filters (Millipore, UFC901096) by centrifugation at 2800 rpm at 4 °C, and was buffer-exchanged into PBS. The protein concentration was measured via a NanoDrop One (Thermo Fisher Scientific), aliquoted, and stored at −80 °C for long-term preservation.

## Phage display library construction and panning

The local ethics committee granted approval for this experiment involving dromedary camel. One dromedary camel was immunized by 2 times subcutaneous injection and 2 times of intramuscular injection, each with 300 μg antigen (DS-Cav1 and DS2), every 3 weeks. Equal volumes of adjuvant were added, with complete Freund's adjuvant (CFA) used for the first time and Freund's incomplete adjuvant used for the rest. Three weeks after the last boost, peripheral blood was collected for serum antibody titer detection by ELISA to evaluate the immunological response. Next, the lymphocytes were isolated from peripheral blood with Ficoll 1.077 (Sigma-Aldrich), and total RNA was extracted from these lymphocytes according to the manufacturer's instructions (OMEGA, IR6834). Reverse transcription was performed using the commercially available reverse transcription system (TAKARA, 6210A). The VHH fragments were amplified from the cDNA template and clone into the phagemid pR2 by homologous recombination. Then, the recombinant plasmids were transformed into TG1 electrocompetent cells by Electroporation System (Bio-Rad, Gene Pulser Xcell™) with the following setting: 2.5 KV, 0.5 ms. The transformants were spread on 15 cm Luria–Bertani (LB) agar plates supplemented with 2% glucose and 50 μg/mL carbenicillin, followed by culture at 37 °C overnight. The colonies were scraped from the plates, added 20 mL of 2×TY medium, mix well, and the phage library was successfully obtained. A portion of the phages was amplified for the panning assay to select specific binders

according to a previous description (Ma et al, 2021; Wang et al, 2025). In brief, microplate was coated with 10 µg of antigen (DS-Cav1) per well and PBS overnight at 4 °C. After blocking, $1 \times 10^{11}$ pfu phage library was added for panning and incubated for 1 h at RT. The wells were washed 30 times with PBST to remove false-positive phages. Next, the positive phages were digested and collected with 0.5 mg/mL trypsin. The monoclonal phages obtained from two rounds of panning were subsequently tested using a phage ELISA. Positive phages showing strong signals in the ELISA were selected for DNA sequencing.

## Expression and purification of antibodies

DNA encoding the candidate Nbs was obtain by PCR and cloned into the PTT5 expression vector. These Nbs were expressed transiently in HEK293F cells as described above and were purified by protein A or NI-NTA affinity column according to the manufacturer's instructions. The positive control antibodies 131-2a (RVV02818), MPE8 (RVV02814), 101F (RVV02816), hRSV90 (RVV02813), and nirsevimab (DVV02802) were all purchased from AntibodySystem. Palivizumab (Absin, abs19040) was purchased from Shanghai Universal Biotech Co., Ltd.

## Enzyme-linked immunosorbent assay (ELISA)

ELISA was used to identify the binding activity of antibodies, and the method of operation was the same as that described previously (Wen et al, 2023). In brief, a 96-well microplate was coated with antigen (DS-Cav1, DS2, or Post-F) at a concentration of 2 µg/mL in PBS and incubated at 4 °C overnight. After washing twice with PBS, the pellet was blocked with 5% milk in PBS for 2 h at room temperature (RT) to prevent non-specific binding. The samples were serially diluted in a 3-fold gradient, with an initial concentration of 30 nM, and 100 µL of each dilution was added to the wells, followed by a 2-h incubation at RT. After washing with PBST, the secondary antibody (rabbit anti-human IgG-Fc HRP; Sino Biological; 10702-T16-H-100, 1:8000) was added and incubated for 1 h. Finally, 3, 3', 5, 5'-Tetramethylbenzidine (TMB) substrate (Beyotime; P0209) was applied for color development, and the reaction was stopped with 0.5 M $H_2SO_4$. The absorbance at 450 nm was measured to determine the level of antibody binding.

## RSV neutralization assay

The neutralizing activity of Nbs against RSV was evaluated using a new high-throughput indirect immunofluorescence assay (Wang et al, 2025; Wen et al, 2019; Xun et al, 2021). Briefly, on the day prior to the experiment, Vero cells were seeded in 96-well plates at a density of $1 \times 10^4$ cells per well. We commenced with a concentration of 100 nM and performed a gradient dilution of the Nbs. RSV A2 and B were mixed with the nanobody dilutions and incubated for 1 h at 37 °C. The mixtures were then added to target cells and incubated at 37 °C for 2 h. After washing once with PBS, Vero cells were cultured in fresh medium containing 2% FBS for 72 h at 37 °C. Finally, cells were fixed with 4% paraformaldehyde (PFA) and stained with the RSV detection antibody F-E2-Fc (1 µg/mL) (Wang et al, 2025) followed by an FITC-conjugated anti-human IgG (Proteintech, SA00003-12, 1:2000). Fluorescence was measured using a high-content imaging system (Moleculardevices,

76177-140) to quantify neutralization by comparing treated wells to virus-only controls.

## Fusion inhibition assay

The ability of the nanobody to inhibit RSV fusion to cells was detected according to described previously (McLellan et al, 2013b; Rossey et al, 2017b). First, Vero cells were seeded in the 96-well plate in advance. Vero cells must be precooled at 4 °C for 1 h and then washed with chilled PBS. RSV A2, RSV B and RSV Long were pre-incubated with Vero cells for 1 h at 4 °C to allow virus attachment. Unbound virus was then washed away with chilled PBS. Then, the serial 3-fold gradient dilutions of Nbs starting at a concentration of 100 nM were added to the target wells, and incubated for 1 h at 4 °C. Next, the plates were cultured at 37 °C for 60–72 h. Finally, the cells were fixed with 4% PFA and analyzed by high-throughput indirect immunofluorescence assay to determine the fluorescence intensity of the Vero cells.

## Competitive enzyme-linked immunosorbent assay

The nanobody cross competition experiment was carried out as described previously (Ma et al, 2021). First, a 96-well microplate was coated with the DS-Cav1 or Post-F at a concentration of 2 µg/mL in PBS and incubated overnight at 4 °C. After blocking with 5% milk in PBS for 2 h at RT. The competitor antibodies (His-tag) were diluted in 5 nM analyte antibody (Fc-tag) solutions, followed by 3-fold serial dilutions, ranging from 10,240 nM to 2.5 nM. Then, the mixture was added to the wells and incubated at RT for 2 h. After washing with PBST, the secondary antibody (Rabbit Anti-Human IgG-Fc HRP; Sino Biological; 10702-T16-H-100) was added and incubated at RT for 1 h. Finally, the steps for the TMB substrate to develop color were exactly the same as the above ELISA.

## Cryo-EM sample preparation and data acquisition

DS-Cav1 protein and Nbs in HEPES buffer (50 mM HEPES, 150 mM NaCl, pH 7.5) were mixed at a molar ration of 1:3.6 and incubated at 4 °C overnight to prepare an antigen-nanobody complex. The final concentration of DS-Cav1 in the mixture was 0.25 mg/mL. Three microliters of the mixture were applied onto glow-discharged R1.2/1.3 Cu 300 mesh holey carbon-coated grid (Quantifoil, Germany). Then, grids were blotted and plunged into liquid ethane using the Vitrobot Mark IV (Thermo Fisher Scientific, Waltham, USA) (Vitrobot settings: chamber humidity 100%, 4 °C, blot force 10, blot time 3.5 s, wait 20 s) and stored in liquid nitrogen.

Cryo-EM data were collected with a CRYO ARM 300 electron microscope (JEOL, Japan) operating at 300 kV equipped with a K3 direct electron detector (Gatan, USA). Parameters for data acquisition were shown in Appendix Table S1. Micrographs were taken and subjected to cryoSPARC (Punjani et al, 2017) for patched motion correction and CTF estimation. Particles were automatically picked by template-picker using templates created from the previously reported structure of DS-Cav1 (PDB: 8DZW) (Wen et al, 2023). After extraction, bad particles were removed based on 2D-classification results. Particle classes showed signatures of nanobody binding were used for heterogenous refinement, homogenous refinement and non-uniform refinement. The yielding

constructs were finally refined with DeepEMhancer (Sanchez-Garcia et al, 2021) using the tightTarget model in cryoSPARC and used for model building and refinement (Appendix Figs. S8 and S9).

## Model building and refinement

The predicted structure of Nbs from Alphafold3 and previously published DS-Cav1 structure (PDB: 8DZW) was used as the initial template for structure refinement. The initial models were refined with Coot (Emsley et al, 2010) manually followed by real-space refinement in Phenix using phenix.real_space_refine (Afonine et al, 2018). Quality of the model refinement was evaluated using MolProbity (Williams et al, 2018) in Phenix.

## Mouse study

Eight-week-old Balb/c mice (weight: 18–21 g) were divided into three groups: prophylactic, therapeutic, and control. Nirsevimab was as positive control, mice were intraperitoneally injected with candidate Nbs (dose: 2 mg/kg, volume: 200 μL) 24 h before or 24 h after challenge with RSV A2. The mouse was lightly anesthetized and was intranasally challenged with $1 \times 10^6$ $TCID_{50}$ of the RSV A2 strain. All the mice were weighed daily and the weights were recorded. Four days later, the mice were sacrificed, and the lung and nasal turbinate tissues were collected for virus titration analysis according to a previously reported protocol (Citation: Dai et al, 2023; Wang et al, 2025; Zhu et al, 2017).

## Determination of virus RNA copies by qRT-PCR

Total RNA was extracted from mouse nasal turbinate and lung tissues using virus RNA reagents 2.0 (Vazyme, RM401), following the manufacturer's instructions. The real-time qPCR experiments were performed on an Applied Biosystems QuantStudio 3 qPCR system by HiScript II One Step qRT–PCR SYBR Green Kit (Vazyme, Q221). RSV N plasmid dilutions ($10^3$ to $10^8$ copies) were used as standard samples to calculate the number of viral RNA copies per tissue. Each sample was tested in triplicate to ensure accuracy. The qRT–PCR sequences of primers were as follows:

Forward primer ACGACGTTGTAAAACGACGGCCAGTG-GATCCTTAGCAAAG
TCAAGTTGAATGATACACTCA.
Reverse primer TTCACACAGGAAACAGCTATGACCAG-GATCCACATACCTATT
AACCCAGTGAATTTATGATTAGCATCT.

## Determination of viral titers

Upon cervical dislocation, lung and nasal turbinate tissues were harvested from mice and homogenized in 1 mL DMEM on ice. The homogenates were centrifuged at 8000 rpm for 10 min to collect the supernatants. The supernatants were subjected to 10-fold serial dilutions and added to Vero cells in 96-well plates for 1 h at 37 °C under 5% $CO_2$. Following incubation, the inoculum was replaced with maintenance medium (DMEM supplemented with 2% FBS), and the plates were returned to the incubator for 3–4 days. Viral infectivity was quantified by determining the median tissue culture

### The Paper Explained

#### Problem

Respiratory syncytial virus (RSV) is a major pathogen responsible for acute lower respiratory tract infections in infants and elderly individuals worldwide, which can lead to severe outcomes such as bronchiolitis, pneumonia, and even death, posing a significant threat to global public health. However, currently available RSV vaccines and antibody-based therapeutics still face limitations in terms of protective efficacy, target populations, or coverage against viral variants. Therefore, there is an urgent need to develop a new generation of RSV-neutralizing antibodies that offer enhanced safety, higher potency, broad-spectrum protection, and cost-effectiveness.

#### Results

We immunized dromedary camel with RSV F protein to establish a phage display library, and screened out a flood of high-affinity, high-neutralization activity nanobodies, among which 1G9 and 1D8 can exhibit specific binding to RSV F and demonstrate effective neutralization against RSV A2, RSV B and RSV Long strains. Moreover, the nanobody 1G9 exhibited remarkable prophylactic and therapeutic efficacy in the mouse model. Structural analyses revealed that 1G9 and 1D8 target a structure pivot site on antigenic site IV of RSV pre-F, which is highly conserved across the RSV A2 and B strains, effectively inhibiting membrane fusion and demonstrating broad-spectrum antiviral activity.

#### Impact

This study presents a structural and functional characterization of 1G9 and 1D8, demonstrating that they are potent broad-spectrum nanobodies targeting RSV. These nanobodies exhibit unique molecular recognition of a conformational pivot site within antigenic site IV, effectively bridging the unstable HRB region and the structurally rigid domain II of the F protein. Both 1G9 and 1D8 effectively inhibited viral replication in a mouse infection model, demonstrating strong prophylactic and therapeutic efficacy—highlighting their significant potential as candidate drugs for the prevention and treatment of RSV.

infective dose ($TCID_{50}$) based on cytopathic effect (CPE) assessment.

## Statistical analysis

All statistical analyses were performed using GraphPad Prism 8.0. Statistical comparisons between groups were made using one-way ANOVA followed by post hoc tests for multiple comparisons, and t tests (and nonparametric tests). A $p$-value of less than 0.05 was considered statistically significant. Statistical significance was shown as $*P < 0.05$, $**P < 0.01$, $***P < 0.001$, and data are shown as the mean ± SEM or mean ± SD.

## Data availability

The datasets produced in this study are available in the following databases: [Cryo-EM structures]: [Protein Data Base] [accession number/identifier 9LM5 and 9LM6].

The source data of this paper are collected in the following database record: biostudies:S-SCDT-10_1038-S44321-026-00412-w.

# Peer review information

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

## Acknowledgements

This work was supported by the Strategic Priority Research Program of the Chinese Academy of Sciences (Grant No. XDB0490000); the National Key R&D Program of China (Grant No. 2022YFC3500804); USTC Research Funds of the Double First-Class Initiative (Grant No. YD9100002040); the Joint Laboratory of Innovation in Life Sciences of USTC and Changchun Zhuoyi Biological Co.Ltd; and the Hubei Provincial Health Research Fund (Grant Number WJ2025Z018).

## Author contributions

Qianqian Wang: Data curation; Formal analysis; Investigation; Methodology; Writing—original draft. Xianliang Ke: Formal analysis; Investigation; Visualization; Writing—original draft. Entao Li: Methodology; Writing—original draft. Dongxiang Hong: Investigation; Methodology. Zekai Cheng: Methodology; Writing—review and editing. Hongxin Li: Methodology. Jiachen Zhang: Methodology. Tengchuan Jin: Methodology. Rui Gong: Methodology. Bo Shu: Conceptualization; Supervision; Funding acquisition; Writing—review and editing. Sandra Chiu: Conceptualization; Supervision; Funding acquisition; Project administration; Writing—review and editing.

Source data underlying figure panels in this paper may have individual authorship assigned. Where available, figure panel/source data authorship is listed in the following database record: biostudies:S-SCDT-10_1038-S44321-026-00412-w.

## Disclosure and competing interests statement

QQW, ETL, DXH, TCJ, and SC are co-inventors on pending patent applications related to the 1G9. The other authors declare no known competing financial interests or personal relationships that could have appeared to influence the work reported in this paper.

