## [Peer Review File · EMBO Molecular Medicine]

Broadly Neutralizing Nanobodies Target a Defined Structural Pivot Site on the RSV Fusion Protein

Qianqian Wang, Xianliang Ke, Entao Li, Dongxiang Hong, Zekai Cheng, Hongxin Li, Jiachen Zhang, Tengchuan Jin, Rui Gong, Bo Shu, and Sandra Chiu

Corresponding author(s): Qianqian Wang (wqq915@ustc.edu.cn) , Bo Shu (shubo@wh.iov.cn), Sandra Chiu (qiux@ustc.edu.cn)

Review Timeline:

Submission Date:	23rd Aug 25
Editorial Decision:	16th Sep 25
Revision Received:	6th Jan 26
Editorial Decision:	21st Jan 26
Revision Received:	19th Feb 26
Accepted:	5th Mar 26

Editor: Zeljko Durdevic

Transaction Report:

16th Sep 2025

Dear Dr. Wang,

Thank you for the submission of your manuscript to EMBO Molecular Medicine. We have now received feedback from the two reviewers who agreed to evaluate your manuscript. As you will see from the reports, both referees are overall supportive of the study but also raise serious concerns that should be addressed in a major revision. Regarding the referee #1 minor point 1, please note that RSV was reclassified as being part of the Pneumoviridae family since 2016. You have indeed provided the correct reference Rima et al, 2017, however in the text you classify RSV as a member of Paramyxoviridae family, please correct. If you would like to discuss further the points raised by the referees, I am available to do so via email or video. Let me know if you are interested in this option.

We would welcome the submission of a revised version within three months for further consideration. Please let us know if you require longer to complete the revision.

I look forward to receiving your revised manuscript.

Yours sincerely,

Zeljko Durdevic

Zeljko Durdevic
Senior Editor
EMBO Molecular Medicine

We require:

- 1) A .docx formatted version of the manuscript text (including legends for main figures, EV figures and tables). Please make sure that the changes are highlighted to be clearly visible.
- 2) Individual production quality figure files as .eps, .tif, .jpg (one file per figure). For guidance, download the 'Figure Guide PDF': (<https://www.embopress.org/page/journal/17574684/authorguide#figureformat>).
- 3) A .docx formatted letter INCLUDING the reviewers' reports and your detailed point-by-point responses to their comments. As part of the EMBO Press transparent editorial process, the point-by-point response is part of the Review Process File (RPF), which will be published alongside your paper.
- 4) A complete author checklist, which you can download from our author guidelines (<https://www.embopress.org/page/journal/17574684/authorguide#submissionofrevisions>). Please insert information in the checklist that is also reflected in the manuscript. The completed author checklist will also be part of the RPF.
- 5) Please note that all corresponding authors are required to supply an ORCID ID for their name upon submission of a revised manuscript.

6) It is mandatory to include a 'Data Availability' section after the Materials and Methods. Before submitting your revision, primary datasets produced in this study need to be deposited in an appropriate public database, and the accession numbers and database listed under 'Data Availability'. Please remember to provide a reviewer password if the datasets are not yet public (see <https://www.embopress.org/page/journal/17574684/authorguide#dataavailability>).

12) Author contributions: You will be asked to provide CRediT (Contributor Role Taxonomy) terms in the submission system. These replace a narrative author contribution section in the manuscript.

13) A Conflict of Interest statement should be provided in the main text.

14) Every published paper now includes a 'Synopsis' to further enhance discoverability. Synopses are displayed on the journal webpage and are freely accessible to all readers. They include a short stand first (maximum of 300 characters, including space) as well as 2-5 one-sentences bullet points that summarizes the paper. Please write the bullet points to summarize the key NEW

findings. They should be designed to be complementary to the abstract - i.e. not repeat the same text. We encourage inclusion of key acronyms and quantitative information (maximum of 30 words / bullet point). Please use the passive voice. Please attach these in a separate file or send them by email, we will incorporate them accordingly.

15) Include a Reagents and Tools Table as part of the Methods section, which can be downloaded from our author guidelines (<https://www.embopress.org/page/journal/17574684/authorguide#structuredmethods>)

**** Reviewer's comments ****

Referee #1 (Comments on Novelty/Model System for Author):

The manuscript entitled "Broadly Neutralizing Nbs Target a Defined Structural Pivot Site on the RSV Fusion Protein" by Wang et al outlines the discovery of broadly neutralizing nanobodies and evaluates their anti-RSV efficacy through in vitro and in vivo experiments. Specificity, 1G9 and 1D8 demonstrated significant neutralizing activity and showed potential for both prophylactic and therapeutic efficacy against RSV infection in mouse models. Notably, 1G9 exhibited superior neutralizing activity against both RSV A2, RSV B and RSV Long strains compared to the existing drug Nirsevimab and significantly reduced viral load in in vivo experiments, indicating a promising application prospect. These findings provide strong scientific evidence for the development of novel prophylactic and therapeutic strategies against RSV.

Referee #1 (Remarks for Author):

The manuscript entitled "Broadly Neutralizing Nbs Target a Defined Structural Pivot Site on the RSV Fusion Protein" by Wang et al outlines the discovery of broadly neutralizing nanobodies and evaluates their anti-RSV efficacy through in vitro and in vivo experiments. Specificity, 1G9 and 1D8 demonstrated significant neutralizing activity and showed potential for both prophylactic and therapeutic efficacy against RSV infection in mouse models. Notably, 1G9 exhibited superior neutralizing activity against both RSV A2, RSV B and RSV Long strains compared to the existing drug Nirsevimab and significantly reduced viral load in in vivo experiments, indicating a promising application prospect. These findings provide strong scientific evidence for the development of novel prophylactic and therapeutic strategies against RSV. It is of high interest to the field and I think the authors did an excellent job. However, the manuscript needs some improvements before publication.

Major comments:

1. In Figure S1A, DS-Cav1 and DS2 should be in a monomer in reduced SDS-PAGE and trimeric protein in non-reduced SDS-PAGE, but the result showed monomer only, yet trimeric protein. Please add the result of non-reduced SDS-PAGE.
2. The Methods section should be improved. please explain how the lung and nasal turbinates were processed to determine the viral titers.
3. This study only evaluated the prophylaxis and treatment of nanobodies against RSV A2 strain, their preventive or therapeutic effects against RSV B strain should also be evaluated.

Minor issues:

1. The author should note that RSV now belongs to the Paramyxoviridae family and the Orthopneumovirus genus.
2. In this study, camel was immunized with RSV F protein antigens such as DS-Cav1 and DS2. The authors should provide a brief introduction in the background.
3. We cannot conclude this statement based on the data from Figure 1 and Figure S2. "Importantly, the candidate Nbs are more effective than nirsevimab in the neutralization of RSV B". Please clarify.
4. In figure3B, "nanobody 1G9 improved the body weight". I think used "maintained body weight" or "attenuated weight loss" would be better.
5. Can the authors provide the sequences of 1G9 and 1D8, so that these don't have to be deduced from the PDB files? The PDB codes of the structures should be provided in the manuscript.

Referee #2 (Remarks for Author):

Review 'Broadly Neutralizing Nbs Target a Defined Structural Pivot Site on the RSV Fusion Protein' by Wang et al.

Wang et al reported on two new nanobodies binding RSV site IV. Despite medium neutralization and binding efficiency, 1G9 does show high protection in vivo. The manuscript needs restructuring and some data needs to be reanalyzed, also the language should be improved. Below you can find my comments (minor and major).

- Abstract: line 33, nirsevimab has made a breakthrough recently. This seems important to report rather than only saying that options are limited.
- Line 53: present 'with'
- Line 60, taking into account the very recent vaccines and antibody prophylaxis regimens taken up in many parts of the world, a downward trend is occurring. Check more recent literature, eg <https://pubmed.ncbi.nlm.nih.gov/40338822/>
- 66, use glycoproteins instead of sugar proteins.
- 59 the F protein IS highly conserved
- 76 Site IV is mostly but not fully conserved in between pre and postfusion conformation.
- 77 'and are sensitive to neutralizing antibodies,' You also have neutralizing abs binding the other antigenic sites (or contacting multiple antigenic sites). But you are correct that antibodies binding preF specific sites are in general more potent
- 97. 'High specificity and affinity' is not a advantage over conventional abs, as this is also the case for full antibodies.
- 103: 'a camel', and was it a camel or a dromedary camel, this is not the same thing (please always specify).
- 120 was this immunization of a mix of the two antigens or consecutive vaccinations?
- Line 121-124: this sentence is very confusing to me, the numbers the OD450 reaches depends also on how long you wait before stopping the reaction. You probably mean the end-point titers? Maybe just mention that there was clear seroconversion (perhaps compare with pre-immunization serum). What does NC stand for in Fig. S1B, please specify in the legend.
- S1D did you compare binding to an irrelevant antigen to rule out aspecific binding?
- Line 131, Please specify how many you expressed.
- Line 132, please show the data from this neut assay.
- 135, please specify why you clone them in a human IgG1 expression vector.
- Line 136, nanobody and nanobody-Fc bands. Always clearly indicate if experiment was done with nanobody or with the Fc fusion.
- Fig S2a, what negative control did you use. Mention the concentration of nanobody/antibody used for detection in the legend.
- What do the numbers in 1E mean? Easier interpretation if you would switch it for a heatmap.
- Binding to site IV also explains the residual binding to post F as this site is not completely changed in the postfusion conformation.
- 166-167: remove the sentence about rsv b neutralization as you did not mention this experiment in this paragraph.
- There is no need to describe the neutralization experiment in that depth in your results section.
- Fig 2, please change x-axis to ng/ml
- Line 180: you cannot compare a nanobody and an antibody when you describe the IC50 in ng/ml. First put it in nM to compensate for the size! Then the antibody will still be more potent.
- Nirsevimab and palivizumab should be able to neutralize RSV B well. This is not in line with previous publications and observations. Is S197N the reason for this reduction in activity? Is it a common sequence feature among rsv B strains? (in clinical strains specifically?)
- You cannot simply state that the nbs prevent the conformational change without proving. Although probably that is the mechanism.
- 1D and 1f, typo in the figure titles.
- 192, again compare in nM. Not expected that palivizumab and nirsevimab perform worse in this type of assay, and not seen in eg <https://www.nature.com/articles/ncomms14158>
- Fig 3a: babl/c is mentioned instead of Balb/c.
- 204: control group instead of model group
- Mention the virus titer the mice received (in PFU). This must have been very high as normally mice do not loose weight after an RSV A2 challenge (with 1.000.000 PFU).
- 202 Say Nb-Fcs if you refer to the fc-fused version (also in your abstract).
- 219: looking at viral titers in the therapeutic set-up is pointless, probably there is still a lot of nanobody or antibody present in the lung sample which has an ex vivo neutralizing effect (because you start from a high dose and extract the lungs 3 days after administration). The qPCR data is more reliable. It is also more custom to look at titers at day 5 post infection for rsv.
- 222, there is no significant difference in the bodyweight data between the groups, you cannot say that there is a mild improvement. The RSV mouse model is not really suitable to show a therapeutic effect, as you need a high titer and there is only very limited replication of the virus.
- It makes more sense to me to discuss the in vivo part in the end of the manuscript. As now, you flip from nanobody to nb-Fc, back to nanobody. It would also make more sense to describe the cryo-EM after the second paragraph as the cryo-em data complements the competition data.
- Line 319 typo: palivizumab
- 320: Palivizumab, missing capital letter
- 331: is this correct, considering IC50 in nM?
- In general language can be improved.
- 338: please specify that this is the Fc-fusion. F-VHH-4 eg was tested in vivo without Fc tail, so hard to compare the two.

- Could you reason why 1G9 was so successful in vivo? I find it hard to understand as it has lower affinity and lower neutralization compared to nirsevimab.
- 392, this would then also be the case for palivizumab, which is less powerful in vivo than nirsevimab. Also by binding preF alone, you can clear virus through ADCC and ADCP.
- Please also report IC50 data from the Nb-Fcs and ELISA data from the Nbs (without fc).
- 407: did the construct you used in the mouse study also contain the YTE mutation? If so please mention clearly.

Response to Reviewers' Comments

Referee #1 (Remarks for Author):

The manuscript entitled "Broadly Neutralizing Nbs Target a Defined Structural Pivot Site on the RSV Fusion Protein" by Wang et al outlines the discovery of broadly neutralizing nanobodies and evaluates their anti-RSV efficacy through in vitro and in vivo experiments. Specificity, 1G9 and 1D8 demonstrated significant neutralizing activity and showed potential for both prophylactic and therapeutic efficacy against RSV infection in mouse models. Notably, 1G9 exhibited superior neutralizing activity against both RSV A2, RSV B and RSV Long strains compared to the existing drug Nirsevimab and significantly reduced viral load in in vivo experiments, indicating a promising application prospect. These findings provide strong scientific evidence for the development of novel prophylactic and therapeutic strategies against RSV. It is of high interest to the field and I think the authors did an excellent job. However, the manuscript needs some improvements before publication.

Response: First, we would like to express our gratitude for your comprehensive summary and positive opinion to our study. We are deeply grateful for the constructive and insightful guidance provided, which has significantly strengthened both the scientific rigor and clarity of our manuscript. In response to the valuable suggestions, we have supplemented our results with additional data and thoroughly revised the article. We sincerely appreciate your contributions to improving our work.

Major comments:

1. In Figure S1A, DS-Cav1 and DS2 should be in a monomer in reduced SDS-PAGE and trimeric protein in non-reduced SDS-PAGE, but the result showed monomer only, yet trimeric protein. Please add the result of non-reduced SDS-PAGE.

Response: We gratefully appreciate this constructive suggestion. Based on your suggestion, we have made the following adjustments:

1) We have included the reduced SDS-PAGE images in Fig.S1A (Response Fig. 1A) to better allow for the assessment of RSV Pre-F (DS-Cav1 and DS2) and Post-F. The monomeric forms of DS Cav1, DS2, and Post-F antigens exhibited molecular weights of ~60 kDa.

2) We have performed the non-reduced SDS-PAGE and Coomassie blue staining for RSV

Pre-F (DS-Cav1 and DS2) and Post-F, their trimeric forms approached ~180 kDa (~60 kDa×3). The trimeric protein may have become unstable after storage or during electrophoresis, resulting in bands of varying sizes (Response Fig. 1B).

3) We also conducted western blot assay in Fig.S1B (Response Fig. 1C) to verify RSV Pre-F (DS-Cav1 and DS2) and Post-F.

Response Fig. 1

2. The Methods section should be improved. please explain how the lung and nasal turbinates were processed to determine the viral titers.

Response: We appreciate your insightful comment and suggestion. We have now included a detailed description of the virus titer assay in our revised manuscript. Our detailed revision is as follows (marked in red; lines 640-647):

“Upon cervical dislocation, lung and nasal turbinate tissues were harvested from mice and homogenized in 1 mL DMEM on ice. The homogenates were centrifuged at 8,000 rpm for 10 minutes to collect the supernatants. The supernatants were subjected to 10-fold serial dilutions and added to Vero cells in 96-well plates for 1 h at 37°C under 5% CO₂. Following incubation, the inoculum was replaced with maintenance medium (DMEM supplemented with 2% FBS), and the plates were returned to the incubator for 3-4 days. Viral infectivity was quantified by determining the median tissue culture infective dose (TCID₅₀) based on cytopathic effect (CPE) assessment.”

3. This study only evaluated the prophylaxis and treatment of nanobodies against RSV A2 strain, their preventive or therapeutic effects against RSV B strain should also be evaluated.

Response: We sincerely appreciate your valuable suggestion. We tried to evaluate the effect of the nanobodies against RSV-B virus in Balb/c mice, the same animal model applied in our RSV A2 study. Mice were intranasally inoculated with 1×10^6 TCID₅₀/mouse in a 100 μ L volume. However, no significant change in body weight was observed following infection (Response Fig. 2), indicating that Balb/c mice exhibit low susceptibility to RSV strain B and therefore are not suitable for nanobody evaluation in this context.

Response Fig. 2

Minor issues:

1. The author should note that RSV now belongs to the Paramyxoviridae family and the Orthopneumovirus genus.

Response: We sincerely appreciate the valuable comment. The recent reports show that the family *Pneumoviridae* comprises large enveloped negative-sense RNA viruses. This taxon was formerly a subfamily within the *Paramyxoviridae*, but was reclassified in 2016 as a family with two genera, *Orthopneumovirus* and *Metapneumovirus* (Rima *et al*, 2017). We have now corrected this statement in the revised manuscript. Our detailed revision is as follows (marked in red; lines 61-62):

“RSV is an enveloped, negative-strand RNA viruses, belonging to the *Pneumoviridae* family and the *Orthopneumovirus* genus.”

2. In this study, camel was immunized with RSV F protein antigens such as DS-Cav1 and DS2. The authors should provide a brief introduction in the background.

Response: We appreciate your comment and suggestion. We have added now the descriptions of two antigens, such as DS-Cav1 and DS2 in the revised manuscript. Our detailed revision is as follows (marked in red; lines 79-86):

“Due to the inherent metastable of the RSV Pre-F protein, its prefusion conformation was stabilized by introducing an intrachain disulfide bond and hydrophobic substitutions via double mutations, yielding the variant designated DS-Cav1. Subsequent iterative optimization of the stability and immunogenicity of DS-Cav1 led to the successful development of the second-generation F protein, DS2. This advancement overcame a major barrier in RSV vaccine and antibody-based therapeutic development, firmly establishing RSV Pre-F as a principal target for both vaccine design and antibody-derived therapies.”

3. We cannot conclude this statement based on the data from Figure 1 and Figure S2. "Importantly, the candidate Nbs are more effective than nirsevimab in the neutralization of RSV B". Please clarify.

Response: Thank you for your insightful suggestion. We apologize for having misplaced this key information in our manuscript, which caused confusion. We have now deleted these inaccurate descriptions.

4. In figure3B, "nanobody 1G9 improved the body weight". I think used "maintained body weight" or "attenuated weight loss" would be better.

Response: We appreciate your insightful comment and suggestion. We agree that the used “maintained body weight” would be better. We have now corrected these inaccurate descriptions in our revised manuscript (lines 330-331).

5. Can the authors provide the sequences of 1G9 and 1D8, so that these don't have to be deduced from the PDB files? The PDB codes of the structures should be provided in the manuscript.

Response: Thank you for this important suggestion. We have provided the sequences of 1G9

and 1D8 data in Appendix Fig. S12 (Response Fig. 3). In addition, we have added PDB codes in the Data availability part of the revised manuscript. Our detailed revision is as follows (marked in red; lines 640-644):

“The cryo-EM structures of the DS-Cav1–1D8 and DS-Cav1–1G9 complexes have been deposited in the Protein Data Bank under accession numbers PDB 9LM5 and PDB 9LM6, respectively. The corresponding cryo-EM electron density maps are available in the Electron Microscopy Data Bank under accession numbers EMD-63216 and EMD-63217.”

Response Fig. 3

	1	10	20	30	40	50	60	70	80	90	100	110	120	128																																																																																									
169	QLQ	VE	SGG	GV	Q	SGG	SLRL	SCA	RS	G	D	T	Y	S	I	Y	S	H	G	N	F	R	Q	P	G	R	E	R	E	G	V	A	I	F	S	D	G	S	T	N	Y	A	S	V	E	G	R	F	T	I	S	Q	D	N	A	R	E	N	T	L	Y	L	Q	N	S	L	K	P	E	T	A	N	Y	F	C	A	R	G	--	N	A	A	H	G	L	V	D	Y	Y	T	E	A	D	F	A	M	G	G	T	Q	T	V	S
108	QLQ	VE	SGG	LV	Q	GG	SLRL	SCA	R	S	E	H	T	Y	C	F	Y	D	H	S	H	R	Q	P	G	K	E	R	E	F	V	S	G	I	D	G	S	A	R	Y	A	D	S	V	K	G	R	F	T	I	S	Q	D	N	A	R	A	N	T	V	Y	L	Q	N	S	L	K	P	E	T	A	N	Y	C	K	T	R	L	C	G	E	A	D	T	Y	C	S	G	G	C	R	D	S	G	Y	M	G	G	T	Q	T	V	S
Consensus	QLQ	VE	SGG	GV	Q	GG	SLRL	SCA	R	S	E	D	T	Y	C	I	Y	D	H	G	N	F	R	Q	P	G	R	E	R	E	G	V	A	I	D	G	S	A	R	Y	A	S	V	E	G	R	F	T	I	S	Q	D	N	A	R	A	N	T	L	Y	L	Q	N	S	L	K	P	E	T	A	N	Y	F	C	A	R		N	A	A	H	G	L	C	D	F	A	M	G	G	T	Q	T	V	S								

Referee #2 (Remarks for Author):

Review 'Broadly Neutralizing Nbs Target a Defined Structural Pivot Site on the RSV Fusion Protein' by Wang et al. Wang et al reported on two new nanobodies binding RSV site IV. Despite medium neutralization and binding efficiency, 1G9 does show high protection in vivo. The manuscript needs restructuring and some data needs to be reanalyzed, also the language should be improved. Below you can find my comments (minor and major).

Response: We gratefully appreciate for your valuable suggestion. Following your suggestions, we have now supplemented several lines of evidence and have corrected several careless mistakes that appeared in our originally submitted manuscript; we have also made some revisions to our manuscript. Our detailed point by point responses are presented below.

1. Abstract: line 33, nirsevimab has made a breakthrough recently. This seems important to report rather than only saying that options are limited.

Response: We appreciate your insightful comment and suggestion. We highly acknowledge the substantial protective effect of the long-acting monoclonal antibody nirsevimab against RSV-associated infection, hospitalization, and severe disease in infants—a truly groundbreaking advance in the field. We have now modified descriptions in Abstract as suggested. Our detailed revision is as follows (marked in red; lines 32-33):

“Although nirsevimab represents a recent breakthrough against RSV infection, the emergence of resistant variants highlights the need for additional antiviral strategies.”

2. Line 53: present 'with'.

Response: Thank you for your insightful suggestion. We have now made corrections in the revised manuscript (marked in red; line 51).

3. Line 60, taking into account the very recent vaccines and antibody prophylaxis regimens taken up in many parts of the world, a downward trend is occurring. Check more recent literature, eg <https://pubmed.ncbi.nlm.nih.gov/40338822/>.

Response: Thank you for pointing out this problem in manuscript. We have now deleted this statement in the revised manuscript.

4. Line 66, use glycoproteins instead of sugar proteins. 59 the F protein IS highly conserved.

Response: We sincerely appreciate the valuable comment. We have made correction as you suggested in the revised manuscript (marked in red; line 64 and line 67).

5. Line 76 Site IV is mostly but not fully conserved in between pre and postfusion conformation.

Response: Thank you for your helpful comment and suggestion. We have now added this statement in the revised manuscript (marked in red; lines 78-79).

6. Line 77 'and are sensitive to neutralizing antibodies,' You also have neutralizing abs binding the other antigenic sites (or contacting multiple antigenic sites). But you are correct that antibodies binding pre-F specific sites are in general more potent.

Response: Thank you for your insightful suggestion. We have deleted “and are sensitive to neutralizing antibodies”, in the revised manuscript (marked in red; line 77).

7. Line 97. 'High specificity and affinity' is not an advantage over conventional abs, as this is also the case for full antibodies.

Response: We thank the reviewer for pointing this out. We agree that high specificity and affinity are not unique advantages of nanobodies compared with conventional antibodies. Our intention was to emphasize that nanobodies possess a structurally distinct paratope, particularly the often-elongated CDR3 loop, which enables them to access recessed or cryptic epitopes that are typically inaccessible to full-length antibodies. We have revised the text accordingly to provide a more accurate description. The updated sentence now reads (line 107):

“Nanobodies (Nbs) have distinct advantages over conventional antibodies: 1) The extended CDR3 loop enables targeting of cryptic epitopes;”

8. Line 103: 'a camel', and was it a camel or a dromedary camel, this is not the same thing (please always specify).

Response: Thank you for your insightful suggestion. The nanobodies used in this study were derived from dromedary camels, and we have updated the manuscript accordingly to ensure

accuracy.

9. Line 120 was this immunization of a mix of the two antigens or consecutive vaccinations?

Response: We are grateful for the suggestion. We immunized dromedary camels with a mixed formulation containing both antigens, DS-Cav1 and DS2. We have clarified this point in the revised manuscript. The revised sentence now reads (lines 113-114):

“In this study, dromedary camel was immunized with RSV F protein antigens (a mixture of DS-Cav1 and DS2)”

10. Line 121-124: this sentence is very confusing to me, the numbers the OD₄₅₀ reaches depends also on how long you wait before stopping the reaction. You probably mean the end-point titers? Maybe just mention that there was clear seroconversion (perhaps compare with pre-immunization serum). What does NC stand for in Fig. S1B, please specify in the legend.

Response: We thank the reviewer for this helpful comment. We agree that the previous description was unclear and that OD₄₅₀ values alone are not an appropriate measure of immune response. Our intention was to indicate that the camel showed clear seroconversion after immunization. To address this, we have simplified the description and now report the antibody titers based on end-point dilution compared with the pre-immune serum (Response Fig.4A-B). We have also clarified in the legend of Appendix Fig. S1 that NC refers to the pre-immunization serum (negative control). The revised text now reads (lines 132-133):

“Clear seroconversion was observed after immunization, as shown by high end-point binding titers compared with pre-immunization serum (Fig. S1).”

Response Fig. 4A-B

11. S1D did you compare binding to an irrelevant antigen to rule out as specific binding?

Response: We sincerely appreciate the valuable comment. To exclude nonspecific binding, we included the phylogenetically related hMPV F protein and five distantly related viral envelope glycoproteins (SFTSV-Gn, NIV-F, EBOV-GP, MARV-GP, MPXV-M1R). The results demonstrated that Nbs 1G9-Fc and 1D8-Fc specifically bound to the RSV F protein, further confirming the specificity of the candidate antibodies (Response Fig.5). These results have been added to in the revised manuscript (Fig. 2F).

Response Fig. 5

12. Line 131, Please specify how many you expressed.

Response: Thank you for your insightful suggestion. Based on sequence diversity and binding affinity, we expressed and purified 55 candidate nanobodies for further screening. These details have been added to the revised manuscript (lines 144-145).

13. Line 132, please show the data from this neut assay.

Response: We appreciate your insightful comment and suggestion. We have now added the result of high-throughput RSV neutralization assay for 55 candidate nanobodies in Appendix Figure S3-4 following your suggestion (Response Fig. 6-7, also shown below).

Response Fig.6

Response Fig.7

14. Line 135, please specify why you clone them in a human IgG1 Fc expression vector.

Response: We thank the reviewer for this suggestion. The nanobodies were cloned into a human IgG1 Fc expression vector to generate homo-bivalent nanobody-Fc fusion proteins. This design extends the in vivo half-life by engaging FcRn (Fang et al, 2025; Rath et al, 2015), enables effector functions such as ADCC and CDC (Foss et al, 2024; Ribeiro et al, 2025; Romano et al, 2018)., and increases avidity through bivalency (De Greve et al, 2020). We have added this clarification in the revised manuscript. The sentence now reads (lines 150-156):

“To further characterize the biological properties of the candidate antibodies, VHH coding regions were cloned into a human IgG1 Fc expression vector, which generates bivalent Fc-fusion nanobodies with extended half-life and effector functions, or into a His-tag expression vector for other in vitro assays (Fig. 1C), and were expressed in HEK293F cells.”

15. Line 136, nanobody and nanobody-Fc bands. Always clearly indicate if experiment was done with nanobody or with the Fc fusion.

Response: We apologize for the omission in the original manuscript. We have now clearly indicated in the revised manuscript whether each experiment was performed with the nanobody alone or with the nanobody-Fc fusion.

16. Fig.S2a, what negative control did you use. Mention the concentration of nanobody/antibody used for detection in the legend.

Response: We are grateful for the suggestion. In the revised Appendix Fig. S5 legend, we have clarified that the negative control in Fig. S2a is PBS (no virus, no antibody). We have also indicated the concentration of the detection nanobodies/antibodies as 10 µg/mL in the figure legend.

17. What do the numbers in 1E mean? Easier interpretation if you would switch it for a heatmap.

Response: We are grateful for the suggestion. We have now switched monoclonal phage ELISA results for a heatmap (minimum value set at 0 and the baseline value at 1) in Fig.1A-B following your suggestion (Response Fig. 11). For instance, nanobody 1A1 denotes the monoclonal phage picked from well A1 of 96-well plate, using DS-Cav1 as the antigen for

panning. Correspondingly, nanobody 2A1 refers to the monoclonal phage isolated from well A1, but selected DS2 as the antigen.

Response Fig.8

18. Binding to site IV also explains the residual binding to post F as this site is not completely changed in the postfusion conformation.

Response: We gratefully appreciate for your valuable suggestion. We have now added this statement in the revised manuscript following your advice (marked in red; lines 78-79).

19. Line 166-167: remove the sentence about RSV B neutralization as you did not mention this experiment in this paragraph.

Response: We are grateful for the suggestion. We have removed this sentence.

20. There is no need to describe the neutralization experiment in that depth in your results section. Fig 2, please change x-axis to ng/ml.

Response: We gratefully appreciate for your valuable suggestion. We have now corrected these redundant descriptions in the Results section and changed the x-axis label to "nM" for neutralization assays in vitro (marked in red; lines 188-199).

21. Line 180: you cannot compare a nanobody and an antibody when you describe the IC_{50} in ng/ml. First put it in nM to compensate for the size! Then the antibody will still be more potent.

Response: We are grateful for the suggestion. We have now corrected the result of neutralization assays and membrane fusion inhibition assay in Fig.3A-F (Response Fig. 9) by

converting the IC₅₀ values of nanobodies and antibodies from ng/mL to nM to account for their size differences.

Response Fig.9

22. Nirsevimab and palivizumab should be able to neutralize RSV B well. This is not in line with previous publications and observations. Is S197N the reason for this reduction in activity? Is it a common sequence feature among RSV B strains? (in clinical strains specifically?)

Response: We are grateful for the suggestion. We agree that nirsevimab and palivizumab effectively neutralize RSV B9320 and RSV B18537 as reported. However, RSV B comprises multiple subtypes, and our study focused on RSV B strain CH93(18)-18. Sequence alignment with RSV B9320 (GenBank: AY353550.1) revealed four amino acid substitutions at site Ø (S197N ; I206T ; E218A ; T234N) in RSV B strain CH93(18)-18, the virus used in our study, which may reduce nirsevimab's neutralizing activity. Notably, in the phase IIb trial, two clinical isolates with substitutions at site Ø exhibited resistance to nirsevimab. While these

mutations have been observed in clinical strains, they are not common(Sun *et al*, 2024).

23. You cannot simply state that the Nbs prevent the conformational change without proving. Although probably that is the mechanism.

Response: We are grateful for this important comment. We agree that the structural transition of F protein cannot be directly proven by our current data. To address this, we performed a membrane fusion inhibition assay, which directly demonstrates that 1G9 and 1D8 can inhibit membrane fusion mediated virus entry. Structural analysis shows that these nanobodies bind to the pivot region of site IV, which is directly involved in the conformational transition of F protein required for fusion. Based on this, we speculate that binding of 1G9 and 1D8 may prevent this structural transition, thereby contributing to their antiviral activity.

24. 1D and 1f, typo in the figure titles.

Response: We are grateful for the suggestion. We have made corrections accordingly.

25. Line 192, again compare in nM. Not expected that palivizumab and nirsevimab perform worse in this type of assay, and not seen in eg<https://www.nature.com/articles/ncomms14158>.

Response: Thank you for your valuable reminder. We have now corrected the result of neutralization assays and membrane fusion inhibition assay in Fig.3A-F (Response Fig. 10-11, also shown below). We have put nanobody and antibody in nM to compensate for the size. At present, there are diverse methods for evaluating the neutralization titers of RSV antibodies including classical viral plaque assay, tissue culture infective dose 50% (TCID₅₀), immunofluorescence (IF) assay, etc. The half-maximal inhibitory concentration (IC₅₀) values of an antibody can vary significantly depending on the assay method used. For example, the IC₅₀ of palivizumab against the RSV A2 strain was measured to be greater than 10 µg/mL in a modified microneutralization assay (Xun *et al*, 2021), whereas it was found to be 137.30 ng/mL in plaque reduction neutralization experiments (Sun *et al.*, 2024).

Response Fig.10 (neutralization assay)

Response Fig.11 (fusion inhibition assay)

26. Fig 3a: babl/c is mentioned instead of Balb/c.

Response: We are grateful for the suggestion. We have made corrections accordingly.

27. Line 204: control group instead of model group

Response: We sincerely appreciate the valuable comment. We have now corrected it in the revised manuscript (marked in red; line 327).

28. Mention the virus titer the mice received (in PFU). This must have been very high as normally mice do not lose weight after an RSV A2 challenge (with 1.000.000 PFU).

Response: We are grateful for this comment. The mice in our study were intranasally inoculated with 1×10^6 PFU of RSV A2. At this dose, we indeed observed significant weight loss, which is consistent with other studies using the same virus strain and mouse model (see References 1–2 below).

[1] Yang Q, Xue B, Liu F, et al. Farnesyltransferase inhibitor lonafarnib suppresses respiratory syncytial virus infection by blocking conformational change of fusion glycoprotein. *Signal Transduct Target Ther.* 2024;9(1):144. Published 2024 Jun 10.

[2] Entao Li, Chendong Yang, Qianqian Wang, Huiling Dong, Zekai Cheng, Weiqi Wang, Xuanjun Wu, Sandra Chiu. (2025). A Self-Assembled RSV-DS2 Based Nanovaccine Elicits a Strong Immune Response Against Respiratory Syncytial Virus Infection. *Advanced Functional Materials*, 28 August 2025.

29. Line 202 Say Nb-Fcs if you refer to the fc-fused version (also in your abstract).

Response: We are grateful for the suggestion. We have corrected the text to refer to the Fc-fused constructs as Nb-Fcs throughout the manuscript, including the Abstract.

30. Line 219: looking at viral titers in the therapeutic set-up is pointless, probably there is still

a lot of nanobody or antibody present in the lung sample which has an ex vivo neutralizing effect (because you start from a high dose and extract the lungs 3 days after administration). The qPCR data is more reliable. It is also more custom to look at titers at day 5 post infection for RSV.

Response: We are grateful for the suggestion. In this study, we evaluated viral burden using both viral titration and qPCR, and both methods consistently showed that the nanobodies exert antiviral effects in vivo.

We agree that the presence of high concentrations of nanobody or antibody in the lung sample may affect ex vivo viral titers. However, measuring viral titers in lung tissue remains a commonly used approach to assess in vivo antiviral protection, as it reflects the overall effect of the antibody on viral replication within the animal. Many previous studies have employed the same method (e.g., refs 1-2 below), and we followed this approach for consistency.

Regarding the suggested time point, many studies reported that lungs and noses were harvested at 4 or 5 days postinfection to assess virus load (Detalle *et al*, 2016; Lianpan Dai, 2023; Tang *et al*, 2019; Zhu *et al*, 2017). In our study, we focused on day 4 to capture early viral dynamics and the immediate protective effect of the administered nanobody.

[1] Sun Y, Liu L, Qiang H, et al. A potent broad-spectrum neutralizing antibody targeting a conserved region of the prefusion RSV F protein. *Nat Commun.* 2024;15(1):10085. Published 2024 Nov 21.

[2] Zhu Q, McLellan JS, Kallewaard NL, et al. A highly potent extended half-life antibody as a potential RSV vaccine surrogate for all infants. *Sci Transl Med.* 2017;9(388):eaaj1928.

31. Line 222, there is no significant difference in the bodyweight data between the groups, you cannot say that there is a mild improvement. The RSV mouse model is not really suitable to show a therapeutic effect, as you need a high titer and there is only very limited replication of the virus.

Response: We are grateful for the suggestion We have now corrected this statement in the revised manuscript. And we have also added relevant discussions in the revised manuscript (marked in red; lines 391-397).

“In the therapeutic setting, we did not observe significant differences in bodyweight among groups. This is likely because a substantial change in viral burden is required to induce measurable weight loss in this mouse model, making bodyweight an insensitive readout for therapeutic assessment. Nevertheless, both viral titration and qPCR analyses demonstrated a clear reduction of viral load in nanobody-treated animals, indicating a robust in vivo antiviral effect despite the absence of detectable weight differences.”

32. It makes more sense to me to discuss the in vivo part in the end of the manuscript. As now, you flip from nanobody to nb-Fc, back to nanobody. It would also make more sense to describe the cryo-EM after the second paragraph as the cryo-EM data complements the competition data.

Response: We gratefully appreciate this constructive suggestion. The section on in vivo animal experiments has been relocated to the end of the manuscript. Additionally, the cryo-EM data is now presented in the subsequent section following the competition ELISA results.

33. Line 319 typo: palivizumab. Line 320: Palivizumab, missing capital letter.

Response: We are grateful for the suggestion. The spelling and capitalization of “palivizumab” have been corrected in the revised manuscript (marked in red; line 355).

34. Line 331: is this correct, considering IC₅₀ in nM? In general language can be improved.

Response: Thank you for your valuable reminder. We have thoroughly reviewed the manuscript and made some corrections.

35. Line 338: please specify that this is the Fc-fusion. F-VHH-4 eg was tested in vivo without Fc tail, so hard to compare the two.

Response: We think this is an excellent suggestion. We have now corrected this statement in the revised manuscript (marked in red; lines 376-380).

36. Could you reason why 1G9 was so successful in vivo? I find it hard to understand as it has lower affinity and lower neutralization compared to nirsevimab.

Response: We are grateful for the suggestion. Although 1G9-Fc shows lower binding affinity and neutralization potency than nirsevimab in vitro, several factors may explain why its in vivo protection appears comparable. First, the in vivo dose was administered on a mg/kg basis. Because nanobody-Fc constructs have a substantially lower molecular weight (~76 kDa) than full-length monoclonal antibodies (~150 kDa), the same mass dose results in a much higher molar amount of 1G9-Fc. The IC₅₀ values of nirsevimab versus 1G9-Fc are 0.11 nM vs 0.47 nM for RSV A2 and 0.37 nM vs 2.05 nM for RSV Long, respectively; under the high in

vivo dosing used in this study, both antibodies were present at concentrations far exceeding their IC₅₀ values. This likely minimized the functional differences observed in vitro and resulted in similar levels of protection in vivo.

37. Line 392, this would then also be the case for palivizumab, which is less powerful in vivo than nirsevimab. Also, by binding pre-F alone, you can clear virus through ADCC and ADCP.

Response: We gratefully appreciate for your valuable suggestion. We agree that the original discussion regarding dual-state binding as the explanation for 1G9/1D8 in vivo efficacy was not fully justified. Therefore, we have removed this part from the manuscript and revised the discussion to focus on more substantiated mechanistic features, including the engagement of the highly conserved pivot region on site IV.

38. Please also report IC₅₀ data from the Nb-Fcs and ELISA data from the Nbs (without Fc).

Response: We gratefully appreciate this constructive suggestion. We have now added IC₅₀ data from the Nbs 1G9-Fc and 1D8-Fc in Fig.3A-C and EC₅₀ data for the Nbs without Fc (1G9-His and 1D8-His) in Fig. 2E following your suggestion.

39. Line 407: did the construct you used in the mouse study also contain the YTE mutation? If so, please mention clearly.

Response: We are grateful for the suggestion. The 1G9-Fc construct used in the Balb/c mouse study did not contain the YTE mutation, and we have clarified this in the revised manuscript.

References

De Greve H, Viridi V, Bakshi S, Depicker A (2020) Simplified monomeric VHH-Fc antibodies provide new opportunities for passive immunization. *Curr Opin Biotechnol* 61: 96-101

Detalle L, Stohr T, Palomo C, Piedra PA, Gilbert BE, Mas V, Millar A, Power UF, Stortelers C, Allosery

K *et al* (2016) Generation and Characterization of ALX-0171, a Potent Novel Therapeutic Nanobody for the Treatment of Respiratory Syncytial Virus Infection. *Antimicrob Agents Chemother* 60: 6-13

Fang Y, Song M, Pu T, Song X, Xu K, Shen P, Cao T, Zhao Y, Hsu S, Han D *et al* (2025) Enhancing the Protein Stability of an Anticancer VHH-Fc Heavy Chain Antibody through Computational Modeling and Variant Design. *Adv Sci (Weinh)* 12: e2500004

Foss S, Sakya SA, Aguinagalde L, Lustig M, Shaughnessy J, Cruz AR, Scheepmaker L, Mathiesen L, Ruso-Julve F, Anthi AK *et al* (2024) Human IgG Fc-engineering for enhanced plasma half-life, mucosal distribution and killing of cancer cells and bacteria. *Nat Commun* 15: 2007

Lianpan Dai JS, Lili Xu, Zhao Gao, Senyu Xu, Yan Chai, Liang Wang, Mi Yang, Tong Ma, Qihui Wang, Sushan Cao, Junming Yie, Gang Zou, Zhengde Xie, Jim Zhen Wu, George Fu Gao (2023) A protective human antibody against respiratory syncytial virus by targeting a prefusion epitope across sites IV and V of the viral fusion glycoprotein. *hLife* 1: 12-25

Rath T, Baker K, Dumont JA, Peters RT, Jiang H, Qiao SW, Lencer WI, Pierce GF, Blumberg RS (2015) Fc-fusion proteins and FcRn: structural insights for longer-lasting and more effective therapeutics. *Crit Rev Biotechnol* 35: 235-254

Ribeiro R, Vitor JMB, Voronovska A, Moreira JN, Goncalves J (2025) Novel Strategy of Antibody Affinity Maturation and Enhancement of Nucleolin-Mediated Antibody-Dependent Cellular Cytotoxicity Against Triple-Negative Breast Cancer. *Biotechnol J* 20: e202400380

Rima B, Collins P, Easton A, Fouchier R, Kurath G, Lamb RA, Lee B, Maisner A, Rota P, Wang L *et al* (2017) ICTV Virus Taxonomy Profile: Pneumoviridae. *J Gen Virol* 98: 2912-2913

Romano S, Moura V, Simoes S, Moreira JN, Goncalves J (2018) Anticancer activity and antibody-dependent cell-mediated cytotoxicity of novel anti-nucleolin antibodies. *Sci Rep* 8: 7450

Sun Y, Liu L, Qiang H, Sun H, Jiang Y, Ren L, Jiang Z, Lei S, Chen L, Wang Y *et al* (2024) A potent broad-spectrum neutralizing antibody targeting a conserved region of the prefusion RSV F protein. *Nat*

Commun 15: 10085

Tang A, Chen Z, Cox KS, Su HP, Callahan C, Fridman A, Zhang L, Patel SB, Cejas PJ, Swoyer R *et al* (2019) A potent broadly neutralizing human RSV antibody targets conserved site IV of the fusion glycoprotein. *Nat Commun* 10: 4153

Xun G, Song X, Hu J, Zhang H, Liu L, Zhang Z, Gong R (2021) Potent Human Single-Domain Antibodies Specific for a Novel Prefusion Epitope of Respiratory Syncytial Virus F Glycoprotein. *J Virol* 95: e0048521

Zhu Q, McLellan JS, Kallewaard NL, Ulbrandt ND, Palaszynski S, Zhang J, Moldt B, Khan A, Svabek C, McAuliffe JM *et al* (2017) A highly potent extended half-life antibody as a potential RSV vaccine surrogate for all infants. *Sci Transl Med* 9

21st Jan 2026

Dear Dr. Wang,

Thank you for the submission of your revised manuscript to EMBO Molecular Medicine. I am pleased to inform you that we will be able to accept your manuscript pending the following final amendments:

- 1) Please implement referee #2 suggestions.
- 2) Source Data: Please upload source data for the main figures as one folder per figure and the rest of source data as one zipped folder.
- 3) In the main manuscript file, please do the following:
 - Please address all comments suggested by our data editors listed below:
 - o Figure legends:
 1. Please note that the exact p values are not provided in the legends of figures 7B-K.
 2. Please note that information related to n is missing in the legends of figures 7J, K.
 - Correct the order of manuscript sections to: Title page - Abstract & Keywords - Introduction - Results - Discussion - Methods - Data Availability - Acknowledgments - Disclosure Statement & Competing Interests - References - Figure Legends - (Main Tables with legends if applicable) - Expanded View Figure Legends.
 - Indicate in legends exact n and exact p values, not a range, along with the statistical test used. To keep the figures "clear" some authors found providing an Appendix table Sx with all exact p-values preferable. You are welcome to do this if you want to.
 - Author contributions: Please remove it from the manuscript and specify author contributions in our submission system. CRediT has replaced the traditional author contributions section because it offers a systematic machine-readable author contributions format that allows for more effective research assessment. You are encouraged to use the free text boxes beneath each contributing author's name to add specific details on the author's contribution. More information is available in our guide to authors:
<https://www.embopress.org/page/journal/17574684/authorguide#authorshipguidelines>
 - Please make sure to provide access to PDB and EMDB entries. Use the following format to report the accession number of your data:

[data type]: [full name of the resource] [accession number/identifier] ([doi or URL or identifiers.org/DATABASE:ACCESSION])

Please check "Author Guidelines" for more information.

<https://www.embopress.org/page/journal/17574684/authorguide#availabilityofpublishedmaterial>

- 4) Appendix: Please upload it as a PDF file, remove author information from the title page and add page numbers to the table of contents.
- 5) The Paper Explained: Please move it to the main manuscript file.
- 6) Synopsis:
 - Synopsis image: Please resize the image to 550 px-wide x 300-600 pixels high.
 - Please check your synopsis text and image before submission with your revised manuscript. Please be aware that in the proof stage minor corrections only are allowed (e.g., typos).
- 7) As part of the EMBO Publications transparent editorial process (see our Editorial at <http://embomolmed.embopress.org/content/2/9/329>), EMBO Molecular Medicine will publish online a Review Process File (RPF) to accompany accepted manuscripts. This file will be published in conjunction with your paper and will include the anonymous referee reports, your point-by-point response and all pertinent correspondence relating to the manuscript. Let us know if you want to remove or not any figures from it prior to publication. Please note that the Authors checklist will be published at the end of the RPF.
- 8) Please provide a point-by-point letter INCLUDING my comments as well as the reviewer's reports and your detailed responses (as Word file).

I look forward to reading a new revised version of your manuscript as soon as possible.

Yours sincerely,

Zeljko Durdevic

Zeljko Durdevic
Senior Editor
EMBO Molecular Medicine

*** Instructions to submit your revised manuscript ***

When preparing your revised manuscript, please refer to our guidelines: <https://link.springer.com/journal/44321/submission-guidelines#cms-Revised-submissions>. We perform an initial quality control of all revised manuscripts before re-review; failure to include requested items will delay the evaluation of your revision.

We require:

- 1) A .docx formatted version of the manuscript text (including legends for main figures, EV figures and tables). Please make sure that the changes are highlighted to be clearly visible.
- 2) Individual production quality figure files as .eps, .tif, .jpg (one file per figure). For guidance, download the 'Figure Guide PDF': <https://media.springernature.com/original/springer-cms/rest/v1/content/27825798/data/v1>.
- 3) A .docx formatted letter INCLUDING the reviewers' reports and your detailed point-by-point responses to their comments. As part of the EMBO Press transparent editorial process, the point-by-point response is part of the Review Process File (RPF), which will be published alongside your paper.
- 4) A complete author checklist, which you can download from our author guidelines. Please insert information in the checklist that is also reflected in the manuscript. The completed author checklist will also be part of the RPF.
- 5) Please note that all corresponding authors are required to supply an ORCID ID for their name upon submission of a revised manuscript.
- 6) It is mandatory to include a 'Data Availability' section after the Materials and Methods. Before submitting your revision, primary datasets produced in this study need to be deposited in an appropriate public database, and the accession numbers and database listed under 'Data Availability'. Please remember to provide a reviewer password if the datasets are not yet public.

7) For data quantification: please specify the name of the statistical test used to generate error bars and P values, the number (n) of independent experiments (specify technical or biological replicates) underlying each data point and the test used to calculate p-values in each figure legend. The figure legends should contain a basic description of n, P and the test applied. Graphs must include a description of the bars and the error bars (s.d., s.e.m.).

9) Our journal encourages inclusion of *data citations in the reference list* to directly cite datasets that were re-used and obtained from public databases. Data citations in the article text are distinct from normal bibliographical citations and should directly link to the database records from which the data can be accessed. In the main text, data citations are formatted as follows: "Data ref: Smith et al, 2001" or "Data ref: NCBI Sequence Read Archive PRJNA342805, 2017". In the Reference list, data citations must be labeled with "[DATASET]". A data reference must provide the database name, accession number/identifiers and a resolvable link to the landing page from which the data can be accessed at the end of the reference.

- the medical issue you are addressing,

- the results obtained and

- their clinical impact.

12) Author contributions: You will be asked to provide CRediT (Contributor Role Taxonomy) terms in the submission system. These replace a narrative author contribution section in the manuscript.

13) A Conflict of Interest statement should be provided in the main text.

14) Every published paper includes a 'Synopsis' to further enhance discoverability. Synopses are displayed on the journal webpage and are freely accessible to all readers. They include a short stand first (maximum of 300 characters, including space) as well as 2-5 one-sentences bullet points that summarizes the paper. Please write the bullet points to summarize the key NEW findings. They should be designed to be complementary to the abstract - i.e. not repeat the same text. We encourage inclusion of key acronyms and quantitative information (maximum of 30 words / bullet point). Please use the passive voice. Please attach these in a separate file or send them by email, we will incorporate them accordingly.

15) Include a Reagents and Tools Table as part of the Methods section, which can be downloaded from our author guidelines.

Photos 400-800 DPI

*Additional important information regarding figures and illustrations can be found at <https://media.springernature.com/original/springer-cms/rest/v1/content/27825798/data/v1>

***** Reviewer's comments *****

Referee #1 (Comments on Novelty/Model System for Author):

The authors used this model system to discover the broadly neutralizing nanobodies 1G9 and 1D8 and evaluate their anti-RSV efficacy through in vitro and in vivo experiments. They demonstrated their significant neutralizing activity and showed potential for both prophylactic and therapeutic efficacy against RSV infection in mouse models.

Referee #1 (Remarks for Author):

The authors have satisfactorily addressed all the reviewers' comments and made the requested revisions, which have significantly improved the manuscript to meet the high standards of EMBO Molecular Medicine. I therefore recommend the paper for immediate acceptance.

Referee #2 (Remarks for Author):

The reviewed version has improved significantly and is almost ready for publication. I would like to point out a few remaining issues:

There are still some language errors or strange sentences (lines 81, 83, 117); Line 928: change 'and' to 'or' as it seems now that the mice got both the prophylactic and the therapeutic treatment.

Figure 7F: the qPCR results after the therapeutic treatment. Although significant, these differences seem very limited. Could you mention the fold reduction in the text so this is clear to the readers (line 346), also in line 404: this is not a clear reduction based on viral RNA, but limited.

Line 387: F-VHH-4 did have lower IC50 than D25 ( nirsevimab), so it did demonstrate superior efficacy on some level.

Response to Reviewers' Comments

1) Please implement referee #2 suggestions.

Done

2) Source Data: Please upload source data for the main figures as one folder per figure and the rest of source data as one zipped folder.

Done

3) In the main manuscript file, please do the following:

- Please address all comments suggested by our data editors listed below:

Figure legends:

1. Please note that the exact p values are not provided in the legends of figures 7B-K.

Done

2. Please note that information related to n is missing in the legends of figures 7J, K.

Done

- Correct the order of manuscript sections to: Title page - Abstract & Keywords - Introduction

- Results - Discussion - Methods - Data Availability - Acknowledgments - Disclosure Statement & Competing Interests - References - Figure Legends - (Main Tables with legends if applicable)

- Expanded View Figure Legends.

- Indicate in legends exact n and exact p values, not a range, along with the statistical test used.

To keep the figures "clear" some authors found providing an Appendix table Sx with all exact p-values preferable. You are welcome to do this if you want to.

Done

- Author contributions: Please remove it from the manuscript and specify author contributions in our submission system. CRediT has replaced the traditional author contributions section because it offers a systematic machine-readable author contributions format that allows for more effective research assessment. You are encouraged to use the free text boxes beneath each

contributing author's name to add specific details on the author's contribution. More information is available in our guide to authors:

<https://www.embopress.org/page/journal/17574684/authorguide#authorshipguidelines>

Done

- Please make sure to provide access to PDB and EMDB entries. Use the following format to report the accession number of your data:

[data type]: [full name of the resource] [accession number/identifier] ([doi or URL or identifiers.org/DATABASE:ACCESSION])

Done

Please check "Author Guidelines" for more information.

<https://www.embopress.org/page/journal/17574684/authorguide#availabilityofpublishedmaterial>

Done

4) Appendix: Please upload it as a PDF file, remove author information from the title page and add page numbers to the table of contents.

Done

5) The Paper Explained: Please move it to the main manuscript file.

Done

6) Synopsis:

- Synopsis image: Please resize the image to 550 px-wide x 300-600 pixels high.

- Please check your synopsis text and image before submission with your revised manuscript.

Please be aware that in the proof stage minor corrections only are allowed (e.g., typos).

Done

7) As part of the EMBO Publications transparent editorial process (see our Editorial at

Referee #1 (Comments on Novelty/Model System for Author):

The authors used this model system to discover the broadly neutralizing nanobodies 1G9 and 1D8 and evaluate their anti-RSV efficacy through in vitro and in vivo experiments. They demonstrated their significant neutralizing activity and showed potential for both prophylactic and therapeutic efficacy against RSV infection in mouse models.

Referee #1 (Remarks for Author):

The authors have satisfactorily addressed all the reviewers' comments and made the requested revisions, which have significantly improved the manuscript to meet the high standards of EMBO Molecular Medicine. I therefore recommend the paper for immediate acceptance.

Response: First, we would like to express our gratitude for your positive feedback on our revised manuscript and for the time and effort dedicated to this further round of review. We are deeply grateful for the constructive and insightful guidance provided, which has significantly strengthened both the scientific rigor and clarity of our manuscript.

Referee #2 (Remarks for Author):

The reviewed version has improved significantly and is almost ready for publication. I would

like to point out a few remaining issues:

Response: We sincerely appreciate your positive feedback and for pointing out remaining mistakes. Following your suggestions, we have now corrected several careless mistakes that appeared in our submitted manuscript. Our detailed point by point responses are presented below.

1. There are still some language errors or strange sentences (lines 81, 83, 117); Line 928: change 'and' to 'or' as it seems now that the mice got both the prophylactic and the therapeutic treatment.

Response: Thank you for your insightful suggestion. We apologize for the language errors or strange sentences in our manuscript, which caused confusion. We have now deleted these inaccurate descriptions in line 81-83. In addition, we have changed the “and” to “or” for figure legends in lines 928 and made corrections in the revised manuscript (marked in red; line 147).
“In this study, dromedary camel was immunized with a mixture of RSV F protein antigens (DS-Cav1 and DS2) and a nanobody phage library was established.”

2. Figure 7F: the qPCR results after the therapeutic treatment. Although significant, these differences seem very limited. Could you mention the fold reduction in the text so this is clear to the readers (line 346), also in line 404: this is not a clear reduction based on viral RNA, but limited.

Response: We are grateful for the suggestion. We have now added the fold reduction section in the revised manuscript following your advice (marked in red; lines 373). Additionally, we have deleted “qPCR analyses” in the revised manuscript (marked in red; line 430).

“In the therapeutic model, RT-qPCR showed that a significant reduction in the amount of viral RNA in the lung tissues and nasal turbinate of nanobody-treated mice compared with PBS group. Specifically, viral RNA levels in the 1G9 group showed ~2.13-fold and ~12.97-fold reductions in lung tissues and nasal turbinate, respectively; 1D8 group showed ~2.68-fold and ~7.75-fold reductions in lung tissues and nasal turbinate, respectively (Fig. 7F-G).”

3. Line 387: F-VHH-4 did have lower IC50 than D25 ( nirsevimab), so it did demonstrate superior efficacy on some level.

Response: We are grateful for the suggestion. We have now corrected this statement in the revised manuscript. (marked in red; lines 414).

“Despite the effort for developing numerous RSV-neutralizing Nbs, such as m35 (His-tag) (Xun et al, 2021), F-VHH-4 (His-tag) (Rossey et al, 2017a) and ALX-0171 (trimeric nanobody) (Detalle et al., 2016), none have yet been approved for clinical use.”

5th Mar 2026

Dear Dr. Wang,

We are pleased to inform you that your manuscript is accepted for publication and is now being sent to our publisher to be included in the next available issue of EMBO Molecular Medicine.

You may qualify for financial assistance for your publication charges - either via a Springer Nature fully open access agreement or an EMBO initiative. Check your eligibility: <https://link.springer.com/journal/44321/how-to-publish-with-us>

Zeljko Durdevic
Senior Editor
EMBO Molecular Medicine

>>> Please note that it is EMBO Molecular Medicine policy for the transcript of the editorial process (containing referee reports and your response letter) to be published as an online supplement to each paper. If you do NOT want this, you will need to inform the Editorial Office via email immediately. More information is available here: <https://link.springer.com/partners/embo-press/editorial-policies#Peer%20review>